

# An improved parameterisation of ozone dry deposition to the ocean and its impact in a global climate-chemistry model

Ashok K. Luhar[1], Ian E. Galbally[1], Matthew T. Woodhouse[1], Marcus Thatcher[1]

[1]CSIRO Oceans and Atmosphere, Aspendale, 3195, Australia

*Correspondence to*: Ashok K. Luhar (ashok.luhar@csiro.au)

**Abstract.** Schemes used to parameterise ozone dry deposition velocity at the oceanic surface mainly differ in terms of how the dominant term of surface resistance is parameterised. We examine three such schemes and test them in a global climate-chemistry model that incorporates meteorological nudging and monthly-varying reactive-gas emissions. The default scheme invokes the commonly used assumption that the water surface resistance is constant. The other two schemes, named the one-

layer and two-layer reactivity schemes, include the simultaneous influence on the water surface resistance of ozone solubility in water, waterside molecular diffusion and turbulent transfer, and a first-order chemical reaction of ozone with dissolved iodide. Unlike the one-layer scheme, the two-layer scheme can indirectly control the degree of interaction between chemical reaction and turbulent transfer through the specification of a surface reactive layer thickness. A comparison is made of the modelled deposition velocity dependencies on sea-surface temperature (SST) and wind speed with recently reported cruise

based observations. The default scheme overestimates the observed deposition velocities by a factor of 2 to 4 when the chemical reaction is slow (e.g. under colder SSTs in the Southern Ocean). The default scheme has almost no temperature, wind-speed and latitudinal variations in contrast with the observations. The one-layer scheme provides noticeably better variations, but it overestimates deposition velocity by a factor of 2 to 3 due to an enhancement of the interaction between chemical reaction and turbulent transfer. The two-layer scheme with a surface reactive layer thickness specification of 2.5

microns, which is approximately equal to the reacto-diffusive length scale of the ozone-iodide reaction, is able to simulate the field measurements most closely, with respect to absolute values as well as SST and wind-speed dependence. The two-layer scheme yields the largest deposition velocities in the tropics. The global oceanic deposition of ozone determined using this scheme is approximately 80 Tg yr$^{-1}$. This amount is 12% of the modelled total global ozone deposition, is almost half of the original oceanic deposition obtained using the default scheme, and corresponds to a 10% decrease in the original estimate

of the total global ozone deposition. The figure of 12% is much lower than the previously reported modelled estimate of oceanic deposition being roughly one third of total deposition. Deposition parameterisation influences the predicted ozone mixing ratios, especially in the Southern Hemisphere. For the latitudes 45–70°S, the two-layer scheme improves the prediction of ozone observed at an altitude of 1 km by 7% and that within the altitude range 1–6 km by 5% compared to the original scheme.





## 1 Introduction

Ozone ($O_3$) is not emitted directly into the atmosphere but is formed in both the stratosphere and troposphere by photochemical reactions involving natural and anthropogenic precursor species. Ozone is an oxidant as well as a precursor to the formation of hydroxyl and hydroperoxyl radicals that play a critical role in the chemical cycles of many trace gases in the

troposphere. The lifetime of ozone in the troposphere is relatively short, being about 22 days, compared to long-lived and globally well-mixed greenhouse gases such as carbon dioxide ($CO_2$). Ozone acts as a greenhouse gas, and adversely impacts human health and plant productivity (Monks et al., 2015). As a greenhouse gas, ozone has the third largest global warming effect after $CO_2$ and methane ($CH_4$), and thus plays an important role in the Earth's climate system. The tropospheric ozone burden is estimated to have increased by about 40% since preindustrial times as a result of increases in the emissions of

ozone precursors (Young et al., 2013). According to the Fifth Assessment Report (AR5) of the United Nations Intergovernmental Panel on Climate Change (IPCC), the anthropogenic component of the radiative forcing due to ozone is estimated to be $0.35 \pm 0.2$ W m$^{-2}$, of which $0.40 \pm 0.2$ W m$^{-2}$ is the tropospheric ozone contribution and $-0.05 \pm 0.1$ W m$^{-2}$ is the stratospheric contribution. Ozone represents about 15% of the total anthropogenic radiative forcing (including aerosols) estimated for the year 2011 (Climate Change, The Physical Science Basis, 2013, IPCC, AR5, Chapter 8, Anthropogenic and

Natural Radiative Forcing).

Aspects of ozone have been investigated in numerous air quality and climate related studies, including its sources and sinks (e.g., Vingarzan, 2004; Wild, 2007; Stevenson et al., 2006, 2013; Young et al., 2013; Monks et al., 2015). In the troposphere, the ozone budget is determined by transport from the stratosphere, deposition at the Earth's surface, and chemical production and loss. Dry deposition is an important sink of ozone (Galbally and Roy, 1980), which influences near-surface

concentration of ozone as well as its lifetime and long range transport. The 'present-day' (about year 2000) total global dry deposition of ozone estimated by the Atmospheric Chemistry and Climate Model Intercomparison Project (ACCMIP) is of the order of $1094 \pm 264$ Tg yr$^{-1}$ which is almost double the flux of ozone from the stratosphere to the troposphere ($477 \pm 96$ Tg yr$^{-1}$) (Climate Change, The Physical Science Basis, 2013, IPCC, AR5, Chapter 8, Anthropogenic and Natural Radiative Forcing, Table 8.1, page 672; Young et al., 2013). Although the average deposition velocity to the ocean is smaller than that

to terrestrial surfaces, the larger coverage of the Earth's surface by the oceans leads to considerable deposition to water— previous modelling studies estimate that about one third of total ozone deposition is to water (Ganzeveld et al., 2009; Hardacre et al., 2015). The deposition process needs to be properly accounted for when modelling ozone chemistry in the atmosphere, irrespective of whether the application is related to air quality or climate chemistry.

Similar to the land surface, the commonly used parameterisation for dry deposition of gases to the ocean surface is to express

deposition velocity ($v_d$), which is the flux of the gas to the surface divided by its concentration in air near the surface, as the linear sum of three resistances (Wesley, 1989):



$$v_d = \frac{1}{r_a + r_b + r_c},$$ (1)

where $r_a$, $r_b$ and $r_c$ are the resistances exerted to the transfer process in three successive top-to-bottom layers: the aerodynamic resistance $r_a$ is the resistance to mixing by turbulent transport in the atmospheric surface layer, the atmospheric viscous (or quasi laminar) sublayer resistance $r_b$ is the resistance to movement across the thin layer (0.1 – 1 mm) of air that

is in direct contact with the surface and not moving with the mean flow of the wind, and the surface resistance $r_c$ is the resistance to uptake by the underlying surface that can be controlled by physical, chemical and/or biological processes depending on the surface type and species of interest.

For ozone deposition to oceanic surfaces, $r_c$ is the dominant resistance and several approaches have been proposed to calculate it. A common approach is to use a constant value, normally $r_c = 2000$ s m⁻¹, based on Wesely's (1989) deposition

parameterisation. Several global chemical transport models use this approach by default, for example, MATCH-MPIC (von Kuhlmann et al., 2003; Ganzeveld et al., 2009), MOZART-4 (Emmons et al., 2010), CAM-chem (Lamarque et al., 2012), GEOS-Chem (Mao et al., 2013), and UKCA (Abraham et al., 2012).

While a constant $r_c$ may be a good first-order estimate of global and long-term average oceanic dry deposition, it does not include any spatial and temporal dependencies of the surface uptake process on any oceanic physical, chemical or biological

properties or processes. The surface ocean acts as a gateway for molecules to enter the atmosphere or ocean medium, and contains complex chemical reactions of inorganic components and dissolved organic matter. Some of these reactions are important sources or sinks of climatically active trace gases. For example, ozone is known to react with a number of dissolved chemical compounds present in the seawater, with the reaction with dissolved iodide being by far the fastest one (Garland et al., 1980; Chang et al., 2004) which needs to be considered. Apart from providing more accurate ozone

predictions, a proper treatment of deposition is also important from the point of view of better accounting for feedback cycles; for example the ozone reaction with sea-surface iodide produces volatile iodine compounds, which may then participate in catalytic ozone destruction cycles in the marine atmosphere (Carpenter et al., 2013).

For the purpose of clarity we describe the layers of the ocean from the surface down as follows. The top 1 μm to 1 mm of the sea surface is termed the sea surface microlayer (SSM) (Carpenter et al., 2015). The physical and chemical properties of the

SSM can be different to those of the bulk water. The SSM may consist of various sublayers or scales depending on the physical or chemical properties being considered. The upper part of the SSM is a region where turbulent mixing is drastically reduced and molecular diffusion dominates, resulting in strong gradients in gas concentrations and other properties. A reacto-diffusive length scale can be defined as $(D/a)^{1/2}$ (Carpenter et al., 2013), where $D$ is the molecular diffusivity



($m^2$ $s^{-1}$) and $a$ is the reaction rate constant ($s^{-1}$), and is typically 3 μm (at ~ 25°C) for the ozone-iodide reaction in seawater. A molecular (or diffusive) sublayer can be defined as the region where ozone molecular diffusivity is greater than turbulent transfer, and is typically 50 μm thick. A viscous (or quasi laminar) sublayer is where viscous processes effectively dissipate the turbulent energy, and is of the order of 1 mm (Fairall et al., 2000). In the surface turbulent layer (~ 10–50 m) below the

viscous sublayer, turbulent processes dominate.

## 2 Mechanistic approaches to determining $r_c$

Most early studies on ozone deposition to seawater either explored its dependence on dynamical factors related to turbulent transfer without considering any chemical reaction (e.g. Galbally and Roy, 1980). Other studies have considered molecular diffusion coupled with chemical reactions (e.g. with iodide) in the seawater without including any effects of turbulent

transfer (e.g. Garland et al., 1980). Using the assumption of Wanninkhof (1992) that the enhancement of deposition due to chemical processes was additive to the turbulent transfer in water, Chang et al. (2004) formulated $1 / r_c$ as a linear sum of two independent terms with the correct asymptotic behaviours: the first term representing the influence of molecular diffusion coupled with chemical reaction (Garland et al., 1980) and the second term representing the influence of wind induced turbulent transfer coupled with a chemical enhancement factor (Liss and Merlivat, 1986).

More recently, by solving a simplified form of the budget equation for mass conservation that include turbulent and molecular transport and a chemical reaction term, Fairall et al. (2007) derived two formulations for $r_c$ that are able to account for the simultaneous effects of oceanic physical and chemical processes (i.e. ozone solubility, molecular diffusion, turbulent transfer and chemical reaction). Their approach signifies a development based on fundamental conservations laws.

Some of the processes affecting deposition act in opposite directions and some are interlinked (e.g. ozone solubility in water

and reaction rate coefficient are a function of sea surface temperature (SST) but in opposite way). Consequently, it has been difficult to properly test a mechanistic deposition scheme against field measurements without coupling it to an atmospheric composition model and then testing the response of the full coupled system that represents the conditions of the measurements. To our knowledge, there has not previously been such a detailed testing in light of newer deposition observations and better modelling capabilities that could lead to potential improvements in deposition parameterisation.

Ganzeveld et al. (2009) explored the response of a free-running global climate-chemistry model to the choice of a constant $r_c$ and the one-layer reactivity parameterisation derived by Fairall et al. (2007) (described later in Section 5) but a detailed testing against deposition data was not reported, presumably because of the lack of suitable data at the time.

Recently Helmig et al. (2012) presented the first ship-borne open-ocean ozone flux measurements and at the same time the most extensive set of these measurements (1700 hours of observations) probably surpassing all previous data. The

measurements covered the Gulf of Mexico, the southern and northern Atlantic, the Southern Ocean, and the eastern Pacific





Ocean. These experiments gave medians of the oceanic ozone deposition velocity ($v_d$) from the five cruises of 0.009 to 0.034 cm s$^{-1}$. The measurements cover the range of 45°N to 50°S and show little wind-speed dependence but a marked sea surface temperature dependence. This dataset provides a unique opportunity to both test oceanic ozone deposition schemes and re-evaluate the ozone deposition rate to the world's oceans.

The aims of this paper are: (a) to examine schemes of ozone dry deposition to the ocean involving: the default constant $r_c$ assumption, the one-layer reactivity model suggested by Fairall et al. (2007) and a development of their two-layer reactivity model, within a global climate-chemistry model, ACCESS-UKCA, incorporating meteorological nudging and monthly-varying reactive-gas emissions, and to compare the results with the ozone deposition data of Helmig et al. (2012). This comparison enables selection of an improved deposition formulation. Then (b) using the best fitting scheme, re-evaluate the

rate of ozone deposition over the ocean and for the combined ocean-land system, and finally (c) with the best fitting scheme, examine its influence on the comparison of modelled ozone compared with global ozone profile observations covering the troposphere.

**3 ACCESS-UKCA chemistry-climate model**

The Australian Community Climate and Earth-System Simulator (ACCESS, see Bi et al., 2013) has been developed for both

climate and numerical weather prediction purposes. The physical atmosphere component of ACCESS is the UK Met Office's Unified Model (MetUM). The UK Chemistry and Aerosol (UKCA, http://www.ukca.ac.uk, described below) atmospheric composition module (at UM vn8.4; see Abraham et al., 2012) is part of ACCESS, and the resulting model is referred to here as ACCESS-UKCA.

Several different chemical schemes are available within UKCA. The configuration used here (at UM vn8.4) combines the

tropospheric chemistry scheme described by O'Connor et al. (2014) and the stratospheric chemistry as described by Morgenstern et al. (2009). The resulting chemistry configuration is known as Chemistry of the Stratosphere and Troposphere (CheST). The tropospheric chemistry scheme accounts for $O_x$, $HO_x$, $NO_x$, $CH_4$ and other volatile organic carbon species (e.g. isoprene). The stratospheric chemistry scheme includes chlorine and bromine chemistry (for ozone depleting substances, ODS), and heterogeneous polar stratospheric cloud chemistry suitable for simulating stratospheric ozone. The Fast-JX

photolysis scheme (Neu et al., 2007) is used in the CheST configuration, and is applicable both in the troposphere and stratosphere. For aerosols, the size-resolved aerosol microphysics scheme GLOMAP-mode, which includes sulfur chemistry, is used (Mann et al., 2010). The total number of reactions is 306 across 86 species. Interactive gas-phase dry and wet deposition processes are included. The chemical reactions together with relevant deposition rates are solved within the ASAD (A Self-contained Atmospheric chemistry coDe) framework.





UKCA is coupled to a radiation scheme (via $O_3$, $CH_4$, $N_2O$, and aerosol direct and indirect effects). Changes in atmospheric composition are thus allowed to impact on model physics, allowing the representation of feedbacks from chemical composition to atmospheric dynamics when operating in a free-running (un-nudged) climate configuration. ACCESS also has a coupled ocean model, but the present configuration used an atmosphere-only set up forced by observed sea-surface

temperature and sea-ice fields.

The atmospheric model domain is global with 85 levels extending from the surface to approximately 85 km, and the horizontal resolution was 1.875° in longitude and 1.25° in latitude (the so called N96L85 configuration).

In order for a free-running global climate model to realistically reproduce the state of the atmosphere, especially for process studies at short timescales and subsequent comparison with measured data, meteorological nudging is often used. Nudging is

a simple data assimilation technique that uses meteorological reanalyses data to relax dynamical variables of a model towards the observed state of the atmosphere at a given time, thus minimizing meteorology as a source of uncertainty in the modelled fields. The variables nudged are horizontal wind components and potential temperature in the free troposphere by using the ERA-Interim reanalyses on pressure levels (Telford et al., 2008; Uhe and Thatcher, 2015).

A global monthly-varying emissions database for reactive gases and aerosols that includes both anthropogenic and natural

components was compiled for ACCESS-UKCA and is described in detail by Woodhouse et al. (2015), with the exception that here we used GFED4s (including small fires) biomass burning emissions data instead of the original ACCMIP database, the former includes inter-annual variability. For methane, nitrous oxide and ozone depleting substances, concentrations are prescribed instead of emissions.

The model has nine surface types (namely broad-leaf trees, needle-leaf trees, C3 and C4 grass, shrub, urban, water, bare soil,

and land ice) and for a particular grid box the three resistances are calculated for each surface type and a corresponding deposition velocity is then computed. The deposition velocity is uniformly distributed to all model levels contained within the atmospheric boundary layer. A first order loss rate is calculated corresponding to each deposition velocity, and a grid-box mean loss rate is obtained by weighting the individual loss rates with the fractions of the surface types present in the grid box. The mean loss rate is used in the species mass conservation equation. Currently there is only one water surface type in

the model, so the same deposition scheme is used for both seawater and freshwater.

### 3.1 Current scheme for ozone deposition to water

In ACCESS-UKCA, the aerodynamic resistance $r_a$ is a function of sea-surface roughness, wind speed and atmospheric stability, and is determined as

$$r_a = \frac{\ln(z_r/z_0) - \psi}{\kappa\, u_*},$$   (2)





where $u_*$ is the friction velocity, $z_r$ is a reference height (= 50 m), $\kappa$ is the von Karman constant (= 0.4) and $\psi$ is a stability correction function. The aerodynamic roughness length $z_0$ for the sea surface is parameterised as the sum of viscous and gravity wave parts (Smith, 1988):

$$z_0 = \frac{0.11 \, \nu_a}{u_*} + \frac{a_c \, u_*^2}{g}, \tag{3}$$

where the kinematic viscosity of air $\nu_a$ a function of surface temperature and air density, $a_c$ is a Charnock coefficient (= 0.016), and $g$ is the acceleration due to gravity.

The resistance $r_b$ is related to molecular properties of the gas as well as atmospheric turbulence through the following relationship (Hicks et al., 1987):

$$r_b = \left(\frac{S_c}{P_r}\right)^{2/3} \frac{1}{\kappa \, u_*}, \tag{4}$$

where $S_c$ is the Schmidt number defined as the ratio of the kinematic viscosity of air to molecular diffusivity of the gas species in air and $P_r$ is the Prandtl number of air (i.e. the ratio of momentum diffusivity to thermal diffusivity ≈ 0.72). The molecular diffusivity of ozone in air is taken as $1.4 \times 10^{-5}$ m$^2$ s$^{-1}$.

The present version of ACCESS-UKCA uses $r_c = 2200$ s m$^{-1}$, which yields a ceiling value for $v_d$ of $1/r_c = 0.0455$ cm s$^{-1}$. This will be the first (or default) model specification for ozone deposition in our study.

## 4 Ozone loss in seawater

Garland et al. (1980) introduced the idea that ozone loss in the ocean was dominated by its reaction with dissolved iodide ions ($O_3 + I^- \rightarrow$ products) within the seawater. This bimolecular reaction is considered as a pseudo first order in several models of ozone loss in seawater because only a very small proportion of iodide is used and its concentration remains almost constant. The first-order rate coefficient $a$ (s$^{-1}$) is equal to the pertinent second-order rate coefficient ($k$) times the iodide concentration $[I^-]$, i.e.

$$a = k.[I^-], \tag{5}$$



where $[I^-]$ is in mole per litre (or molar, M) and $k$ is in M$^{-1}$ s$^{-1}$. If there are other first order reactions, then the total reaction rate can be determined as $a = \sum_i k_i C_i$, where $k_i$ and $C_i$ are the first order rate coefficient and concentration of the $i^{th}$ species.

Eq. (5) actually represents the integrated reactivity given for the bulk reaction ($O_3 + I^- \rightarrow$ products) and is used to calculate the ozone removal by reaction with iodide. However, it is noted that the chemistry of ozone reaction with iodide in aqueous solution is complex. The following are the suggested reaction steps (Sakamoto et al., 2009):

$$O_3 + I^- \rightarrow I\text{-}OOO^- \tag{6}$$

$$I\text{-}OOO^- \rightarrow IO^- + O_2 \tag{7}$$

$$IO^- + H^+ \rightarrow HOI \tag{8}$$

$$HOI + I^- + H^+ \rightleftarrows I_2 + H_2O \tag{9}$$

$$I_2 + I^- \rightleftarrows I_3^- \tag{10}$$

There can also be complicating factors including reactions with other halogens in seawater (Sarwar et al., 2015), the modification of bulk properties of seawater in the surface microlayer, and the presence of organic compounds in seawater including those preferentially concentrated in the sea surface microlayer (Carpenter et al., 2015). These are not addressed here.

The second-order rate coefficient ($k$) in Eq. (5) is derived from the data of Magi et al. (1997) (see Section 5.1).

### 5 An alternative scheme for $r_c$ in ACCESS-UKCA

Atmospheric ozone in passing into the ocean is transferred by molecular diffusion through the sea-surface microlayer (including viscous sublayer) and then by turbulent processes in the surface turbulent layer. At the same time ozone can be lost by chemical reaction.

Considering current knowledge, assuming that iodide does provide the major sink for ozone in the ocean, the resistance to ozone deposition by the ocean surface will be an unknown function of a number of known variables:

$$r_c = f([I^-], k, T_s, \alpha, D, u_{*_w}), \tag{11}$$




where $\alpha$ is the dimensionless solubility of ozone in water (which is the ratio of the aqueous-phase ozone concentration to its gas-phase concentration and is related to Henry's law coefficient), $T_s$ is the water temperature, $D$ is the molecular diffusivity of ozone in seawater, and $u_{*w}$ is the waterside friction velocity. $r_c$ is expressed as (Liss and Merlivat, 1986):

$$r_c = \frac{1}{\alpha v_{dw}},\tag{12}$$

where $v_{dw}$ is the waterside deposition velocity of ozone. $u_{*w}$ is calculated as $u_{*w} = (\rho_a / \rho_w)^{1/2} u_*$ by assuming that the atmospheric surface stress is equal to the waterside surface stress, where $\rho_a$ is the air density and $\rho_w$ is the water density.

Applying horizontal homogeneity and stationarity to the fundamental equation for the conservation of mass of a reacting and diffusing substance in water yields the following equation in the vertical dimension ($z$, i.e. depth from the water surface) (Fairall et al., 2007)

$$\frac{\partial}{\partial z}\left[(D + K_t)\frac{\partial C}{\partial z}\right] - a\,C = 0,\tag{13}$$

Where $C$ is concentration and $K_t$ is the turbulent diffusivity. By assuming $K_t = \kappa u_{*w} z$, Fairall et al. (2007) solved Eq. (13) to obtain

$$v_{dw} = (a\,D)^{1/2}\left[\frac{K_1(\xi_0)}{K_0(\xi_0)}\right],\tag{14}$$

where $K_n$ is modified Bessel functions of order $n$, $\xi_0 = b(a\,D)^{1/2}$ and $b = 2/(\kappa u_{*w})$. Eq. (14) explicitly includes the role of waterside molecular diffusion, turbulent transfer and first-order chemical reaction. In the limit of fast reaction or negligible turbulence, it reduces to

$$v_{dw} = (a\,D)^{1/2},\tag{15}$$

which is the form proposed by Garland et al. (1980). The above scheme forms part of the COAREG group of gas transfer algorithms (Fairall et al., 2011).

There are four controlling parameters involved in the above scheme for $r_c$ (Eqs. (12) and (14)): ozone solubility, molecular diffusion, turbulent transfer and chemical reaction. Deposition velocity increases as these parameters attain larger values. This scheme is termed the one-layer reactivity scheme due to the assumption of uniform reactivity ($a$) through the ocean depth. We incorporated the above scheme for $r_c$ in ACCESS-UKCA. This will be the second model specification for ozone deposition.





### 5.1 Parameterisation of input quantities

The second-order rate coefficient $k$ (M$^{-1}$ s$^{-1}$) is derived by fitting an Arrhenius-type expression for $k$ as a function of water temperature $T_s$ (K) to the data of Magi et al. (1997):

$$k = \exp\left(\frac{-8772.2}{T_s} + 51.5\right). \tag{16}$$

Ganzeveld et al. (2009) used oceanic surface nitrate concertation as a proxy for iodide concentration. Chance et al. (2014) compiled available measurements of sea-surface iodide which show highest iodide concentrations in tropical waters. They examined statistical relationships between iodide and parameters such as sea surface temperature (SST), nitrate, salinity, chlorophyll-a and mixed layer depth, and found that SST ($T_s$) was the strongest predictor of iodide followed by latitude. MacDonald at al. (2014) used data from several cruises in the Atlantic and Pacific oceans covering the latitudes 50°S to 50°N to derive the following parameterisation for iodide concentration:

$$[I^-](nM) = 1.46 \times 10^{15} \exp\left(\frac{-9134}{T_s}\right). \tag{17}$$

In their study on iodine's impact on tropospheric oxidants, Sherwen et al. (2016) found that the MacDonald at al. (2014) parameterisation provided better results than the iodide-SST regression parameterisation derived by Chance at al. (2014). We use Eq. (17) in our ACCESS-UKCA modelling, but a sensitivity to the iodide parameterisation derived by Chance at al. (2014) will also be explored later. Based on Eq. (17), the concentrations of iodide lie in the range 6 to 160 $nM$, and the reaction time-scale $a$ lies in the range 2 to 1350 s$^{-1}$ for the experimental conditions modelled. Iodide concentrations are highest in warm tropical waters and lowest in cool waters at higher latitudes.

The dimensionless solubility of ozone in water $\alpha$ decreases with water temperature, and is parameterised as (Morris, 1988):

$$\log_{10}(\alpha) = -0.25 - 0.013(T_s - 273.16). \tag{18}$$

The molecular diffusivity $D$ (m$^2$ s$^{-1}$) of ozone in water is expressed as (Johnson and Davis, 1996):

$$D = 1.1 \times 10^{-6} \exp\left(\frac{-1896}{T_s}\right). \tag{19}$$

We ran ACCESS-UKCA with both the default scheme and the above one-layer reactivity scheme. The modified Bessel functions were determined using the numerical algorithms given by Press et al. (1992). The values of $(a\,D)^{1/2}$ (see Eq. (15)) increase by a factor of 30 from high latitude cold waters to tropical warm waters in the model calculations.





**6 Comparison of modelled deposition velocity with data**

Helmig et al. (2012) reported the magnitude and variability of ozone deposition velocity measured over the open ocean from a ship-based eddy-covariance ozone flux system during 2006–2008. The experiments were conducted on five cruises, namely: (1) TexAQS06 (July 7 to September 12, 2006), (2) STRATUS06 (October 9–27, 2006), (3) GOMECC07 (July 11 to

August 4, 2007), (4) GasEx08 (February 29 to April 11, 2008), and (5) AMMA08 cruises (April 27 to May 18, 2008). The respective areas covered were: (1) North-western Gulf of Mexico, (2) the persistent stratus cloud region off Chile in the eastern Pacific Ocean, (3) the Gulf of Mexico and the US east coast, (4) the Southern Ocean, and (5) the southern and northern Atlantic Ocean.

Helmig et al. (2012) plotted the observed $v_d$ values as a function of SST, but in order to compare the data with the $v_d$ values

obtained from the offline one-layer reactivity scheme of Fairall et al. (2007) in which the influence of not all dominant input parameters (e.g. wind speed, reaction rate and oceanic turbulence) could be examined concurrently they removed the wind-speed dependence from their data. This resulted in the scheme not being tested completely for the specific field locations. In the present case, this scheme has been coupled to an atmospheric chemistry model in which all input parameters are supplied and act simultaneously in determining $v_d$ for the specific field program locations and durations, so the modelled value and

data can be compared directly. We obtained the same $v_d$–SST data but without the wind speed dependence removed (Ludovic Bariteau, personal communication, 2016) for a consistent comparison with our model results (although there is not a large difference between the data with or without the wind-speed dependence).

The ACCESS-UKCA model was run from January 2005 until the end of 2008. The model output was used for 2006 onwards and consists of monthly-averaged values of deposition parameters at each grid point. The simulation months and years were

matched with the experimental periods. The model output was extracted at a series of grid-point locations with almost uniform spacings along the tracks of the above experimental cruises, and the modelled values at these points were used for comparison with the measurements. It is noted that the deposition velocity data and modelled values cannot be matched exactly in terms of time and location. The cruise data vary in both space and time and they are averaged with respect to SST or wind-speed bins over a given experimental period. On the other hand, the model values are monthly averages extracted

along the cruise tracks, with the model months matched to the months corresponding to the experimental periods, which means the binning is dominated by spatial changes. This matching difference is not expected to pose significant problem for comparison purposes because the deposition velocity variations are mainly examined in terms of SST which is not expected to vary greatly with time at a given location for a typical experimental period under consideration.

The $v_d$–SST plot in Figure 1 shows that although there are large fluctuations in the data from the various field experiments, a

trend of $v_d$ increasing with SST is apparent. The lowest deposition velocities are observed in the Southern Ocean (GasEx08) and the largest from the TexAQS06 experiment in the North-western Gulf of Mexico. An interpretation of this observed variation is that despite the stronger wind and wave conditions normally present at Southern Ocean latitudes and the higher





ozone solubility due to cooler temperatures there, the deposition velocities are lower because of the lower iodide concentrations at such temperatures which results in a slower ozone-iodide reaction. On the other hand, the observed deposition velocities in the North-western Gulf of Mexico are larger due to warmer waters leading to faster ozone reaction with iodide despite relatively weak winds and lower ozone solubility there.

Figure 1 shows the ACCESS-UKCA predictions obtained using the default (i.e. constant $r_c$) deposition scheme for the five experiments (diamonds – same colour as the data). The range of SSTs from the model for different experiments is in agreement with the measurements, with the lowest values predicted for GasEx08 and the highest for TexAQS06/GOMECC07. However, the modelled $v_d$ is virtually constant at 0.038 cm s$^{-1}$, and the observed variation is progressively overestimated by a factor of 2 to 4 as water temperatures get cooler. This model behaviour with SST is a

consequence of the model using a constant value for the dominant resistance $r_c$, resulting in a near-constant $v_d$ with a ceiling value of 0.0455 cm s$^{-1}$. The difference between the ceiling value and a modelled level corresponds to the contribution made by the airside resistances ($r_a$ and $r_b$ in Eq. (1)) to lower the deposition velocity. The comparison in Figure 1 demonstrates that the constant $r_c$ assumption is not satisfactory both with respect to its absolute value and its lack of variation with SST.

Figure 2 shows the same $v_d$-SST plot as in Figure 1 but with the modelled values obtained using the one-layer reactivity

scheme. Even though the molecular diffusion controlled deposition component, $(a\,D)^{1/2}$, increases by a factor of 30 from high latitude cold waters to tropical warm waters, a variation to that extent is not evident in Figure 2 because turbulent transfer starts to dominate deposition as surface temperatures decrease (this is illustrated in the sensitivity analysis presented later in Section 6.1.1). Compared to Figure 1, the one-layer deposition scheme is able to reproduce the observed trend of $v_d$ increasing with SST noticeably better, accompanied by larger fluctuations in $v_d$, but the model overestimates $v_d$ by a factor of

2 to 3 with the model-data differences getting progressively larger as the water temperatures get cooler.

In Figure 3, the modelled variation of $v_d$ with wind speed (at 10 m height) obtained using the default deposition scheme does not agree with the measurements, with the modelled $v_d$ values almost constant as in Figure 1, whereas the data show an increasing $v_d$ with wind speed for TexAQS06 and to a lesser extent for GOMECC07. For the other three experiments the variation is almost constant, but is much lower in magnitude than that estimated by the model. It is apparent that the higher-

end range of the modelled wind speed is smaller than that of the data, largely because the wind extremes are averaged out in the modelled monthly averages. This not as much an issue in the earlier comparison involving SST because this parameter does not change as much as wind speed temporally. Hence, in the present case, the $v_d$-wind speed comparison is perhaps not as robust as the $v_d$-SST comparison.

Figure 4 compares the measured $v_d$ variations with wind speed with those computed using the one-layer deposition scheme.

The model is able to perform somewhat better than that in Figure 3 for constant $r_c$, with a hint of the model curves for the five experiments separating out in a manner that is qualitatively consistent with the data.





It is apparent from the above comparison that the one-layer mechanistic scheme given by Eq. (14) is able to yield trends in deposition velocity in the right direction, but the observed values are still overestimated overall.

The likely reason for the overestimation of deposition velocity by the one-layer reactivity scheme is that it overestimates the influence of waterside turbulence and, consequently, its interaction with chemical reaction as a results of the use of a

turbulent diffusivity parameterisation $K_t = \kappa u_{*w} z_w$ that is linear in $z_w$, the distance (or depth) from the water surface. This parameterisation is valid in the surface layer with fully developed turbulence, but in the small viscous sublayer ( $z_w < \delta_u$) whose thickness $\delta_u = 11\nu / u_{*w}$ where $\nu$ is the kinematic viscosity of seawater, is of the order of 1 mm under typical conditions, the above $K_t$ parameterisation would overestimate turbulent transfer because it ignores the dissipation of small-scale turbulence by viscosity. This is important because the length scale of the ozone-iodide reaction falls within the

viscous sublayer. It is suggested that $K_t$ varies as $z_w^{m+1}$ in the viscous sublayer, where $m = 2$ for a smooth surface (Fairall et al., 2000). A form for $K_t$ that is valid for both $z_w < \delta_u$ and $z_w > \delta_u$, such as $K_t = \kappa u_{*w} z_w /[1 + (\delta_u / z_w)^m]$, may be considered in Eq. (13) but an analytical solution is currently elusive. Based on this more general form for $K_t$, we can determine that for $z_w << \delta_u$ the form $K_t = \kappa u_{*w} z_w$ would overestimate $K_t$ by a factor of $(\delta_u / z_w)^m$. A consequence of this would be to artificially elevate ozone deposition rates to the ocean, especially in regions where the ozone removal

chemistry in the surface layer is slow (i.e. areas with cooler SSTs).

### 6.1 Two-layer reactivity scheme – improved formulation

Fairall et al. (2007) extended the one-layer scheme by considering a two-layer approach in which a thin layer of depth $\delta_m$ below the water surface has a high reactivity ($a$) and the region below $\delta_m$ has a very small background reactivity ($a_0$) that is present everywhere including in the first layer (so the total reactivity of the first layer is $a + a_0$). The assumption is that

virtually all the chemical reaction takes place within this thin layer which is roughly identified with the surface microlayer.

At this stage we acknowledge that there is iodide below the surface microlayer and that ozone may transfer into that layer and be rapidly destroyed. Thus the two-layer scheme has an arbitrary constraint that differs somewhat from nature. We describe this model as a semi-empirical model where the appropriate physical and chemical variables have been included in the model, where the turbulence scheme has limitations already discussed and whereby an arbitrary condition, the depth of

the reactive layer is imposed. The usefulness of the model depends on two issues: 1) is the model able to accurately represent the available ozone seawater deposition data, and 2) is the model formulation suitable for efficient use within global climate composition models? These will be both addressed.

The expression for the waterside transfer velocity from the two-layer water reactivity model is:




$$v_{dw} = [(a+a_0)D]^{1/2} \left[ \frac{-A_1 I_1(\xi_0) + B_1 K_1(\xi_0)}{A_1 I_0(\xi_0) + B_1 K_0(\xi_0)} \right],$$  (20)

where $I_n$ is modified Bessel function of order $n$, $\xi_0 = b[(a+a_0)D]^{1/2}$ and $b = 2/(\kappa u_*)$. The coefficients $A_1$ and $B_1$ are determined by imposing three boundary conditions: the flux at the water surface obtained using the solution of the mass conservation Eq. (13) should be equal to a constant specified value $F_0$; the concentration at the interface of the two layers should be continuous; and the flux at the interface of the two layers should be continuous. These boundary conditions lead to the following equations, respectively:

$$-A_1 \xi_0 I_1(\xi_0) + B_1 \xi_0 K_1(\xi_0) = \frac{2F_0}{\kappa u_*},$$  (21)

$$A_1 I_0(\xi_{\delta 1}) + B_1 K_0(\xi_{\delta 1}) - B_2 K_0(\xi_{\delta 2}) = 0,$$  (22)

$$-A_1 \xi_{\delta 1} I_1(\xi_\delta 1) + B_1 \xi_{\delta 1} K_1(\xi_{\delta 1}) - B_2 \xi_{\delta 2} K_1(\xi_{\delta 2}) = 0,$$  (23)

where $\xi_{\delta 1} = [2b(a+a_0)(\delta_m + (bD/2))]^{1/2}$ and $\xi_{\delta 2} = [2ba_0(\delta_m + (bD/2))]^{1/2}$. Although Fairall et al. (2007) allude to the condition corresponding to Eq. (23) they used a different equation in lieu of Eq. (23) by imposing the condition that the total depletion of concentration caused by diffusion and reaction through the depth of the medium must equal the surface flux $F_0$. However, their equation involves integration terms that need to be determined numerically, thus increasing the computational time considerably when used in a large-scale atmospheric model such as ours. We found that the results obtained using both the equations are the same. In Eq. (21), $F_0$ can be taken as unity because it cancels out in the formula for deposition velocity expressed as flux divided by concentration at the surface.

To solve Eqs. (21)–(23) for the three unknowns $A_1$, $B_1$ and $B_2$, the expression for $A_1$ obtained from Eq. (21) is substituted in Eqs. (22) and (23), the last two two are then solved for $B_1$ and $B_2$ analytically by inverting the 2×2 matrix of the coefficients $B_1$ and $B_2$. $A_1$ is obtained by putting the solution for $B_1$ back in Eq. (21). The two-layer scheme has not been previously tested with any measurements.

### 6.1.1 Sensitivity analysis

Deposition velocity is sensitive to the value of $\delta_m$ in the two-layer scheme. To explore this, we tested this scheme in stand-alone mode (i.e. outside ACCESS-UKCA). The oceanic component of dry deposition velocity, i.e. $\alpha v_{dw}$ $(=1/r_c)$ was calculated using Eq. (20) as a function of sea surface temperature and reactivity for a range of $\delta_m$ values for a typical value of the waterside friction velocity ($u_{*_w}$) of 0.01 m s$^{-1}$ (which is equivalent to the airside $u_* \approx 0.3$ m s$^{-1}$). The





parameterisations of $[I^-]$, $k$, $D$ and $\alpha$ specified earlier and $a_0 = 10^{-4}$ s$^{-1}$ (Fairall et al., 2007) were used. The results are presented in Figure 5, along with the curves obtained using the one-layer scheme with and without waterside turbulence (i.e. $K_t = 0$). These results show that: 1) the variation of $\alpha v_{dw}$ with SST is very similar to that with reactivity (the latter on logarithmic scale), 2) the thicker the $\delta_m$ the higher the deposition velocity and that for a sufficiently large $\delta_m$ the two-layer

solution approaches the one-layer solution, 3) turning off waterside turbulence in the one-layer scheme, which leads to the limiting behaviour $\alpha v_{dw} = \alpha (a\, D)^{1/2}$, makes the deposition velocity for the one-layer scheme without turbulence diverge further and further to become a very small fraction of the dry deposition velocity for the one-layer scheme with turbulence as SST or reactivity gets smaller, 4) except for cooler SSTs or lower reactivity, the two-layer scheme results in deposition velocity values that are even smaller than those from the one-layer scheme without turbulence for smaller values of $\delta_m$ (such

$\delta_m$ values are obviously not realistic because the no-turbulence behaviour is the lower limit), and 5) at the lower end of SST or reactivity values, there is a slight increase in $\alpha v_{dw}$ with decreasing SST or reactivity, which is mostly due to the larger influence of solubility which increases with decreasing temperature.

Sensitivity of $v_{dw}$ to oceanic iodide concentration can also be explored. Based on their compilation of available measurements of sea-surface iodide covering latitudes 69°S to 66°N, Chance at al. (2014) suggest the following

parameterisation for iodide concentration:

$$[I^-](nM) = 0.225\,(T_s - 273.16)^2 + 19 . \tag{24}$$

The iodide concentrations and also the reactivity values ($a$) determined using Eq. (24) are higher by 207, 190, 154, 84 and 51% compared to Eq. (17) derived by MacDonald at al. (2014) for the SSTs of 5, 10, 20, 30 and 35°C, respectively. Comparison of the curves in Figure 6 obtained using Eq. (24) with those in Figure 5a shows, as expected, that the values of

$\alpha v_{dw}$ are all higher in the former due to higher iodide levels: the one-layer curve is higher by 18, 21, 27, 24 and 17%, the no-turbulence curve is higher by 75, 70, 59, 36 and 23%, and, as an example, the two-layer curve for $\delta_m = 2$ μm is higher by 6, 15, 60, 48 and 25% for the above temperatures, respectively. The two-layer curves from the two iodide concentration formulae require different $\delta_m$ values to be similar: for example, the curve for $\delta_m = 2$ μm in Figure 6 almost coincides with that for $\delta_m = 5$ μm in Figure 5a. We use the parameterisation Eq. (17) for iodide concentration in all subsequent

calculations. Nevertheless, it is clear that the use of Eq. (24) in the one-layer scheme would not describe the data in Figure 2 properly.



Other sensitivity results show that as $u_{*_w}$ gets larger it requires larger values of $\delta_m$ to approach the one-layer behaviour, and that the solution obtained by the two-layer model approaches the one-layer model when $a_0 \rightarrow a$.

### 6.1.2 Comparison with data

The two-layer reactivity scheme was incorporated into ACCESS-UKCA. The value of $\delta_m$ is selected based on the best

agreement with data, given the constraint that it cannot be smaller than the reacto-diffusive length scale over which chemical reaction controls deposition. In other words, the variation of deposition yielded by a selected $\delta_m$ cannot be lower than the no turbulence case (i.e., $K_t = 0$) for the same input conditions (see, for example, Figure 5). Based on the above considerations, a value of $\delta_m = 2-3$ μm was found to be appropriate (which is of the order of the reacto-diffusive length scale) and we selected $\delta_m = 2.5$ μm. The model variation of $v_d$ with SST determined using the two-layer reactivity scheme is presented in

Figure 7 along with the observed data. There is now a substantial improvement in the prediction of the observed $v_d$ compared to the results from the default and one-layer schemes presented earlier. The modelled variation passes almost through the middle of the data. The variation of $v_d$ with wind speed in Figure 8 also shows considerable improvement over the previous model results, but as mentioned earlier the higher-end range of the modelled wind speed is smaller due to the fact that the wind extremes are averaged out in the modelled monthly averages. There is a relatively large scatter in the GOMECC07

predictions, but the modelled values for the rest of the experiments are similar to the measurements. These results imply that the two-layer scheme has got the combined influences of solubility, molecular diffusion, chemical reaction and turbulent transfer into better balance than the other two schemes.

The reason the two-layer scheme works is as follows. As mentioned earlier, there is an overestimation of the interaction between the chemical reaction term ($a$) and the total diffusivity term $(D + K_t)$ in Eq. (13) within the viscous sublayer

because of the overestimation of $K_t$ as a result of the assumption of its linearly increasing variation with depth. This leads to larger values of deposition velocity than observed. In the two-layer scheme, we effectively compensate for this overestimation of turbulent transfer by constraining the extent of chemical reaction by limiting the iodide reactivity to a thickness of $\delta_m$ equivalent to the reacto-diffusive length scale even though in reality iodide is present through the depth of the oceanic surface layer.

We also speculate that there may be some overestimate in the current reaction pathway for ozone within the ocean. Most of the chemistry underlying the kinetics has been studied at iodide concentrations varying from mM to M, whereas in the ocean the iodide concentration is approximately a million times more dilute, being in the nM range. The fundamental step by step chemistry of ozone reaction with iodide has not been fully characterized. For example if the first step yields $I\text{-}OOO^-$ in





Eq. (6) and the bonding of this complex is weak, there could be a reverse reaction freeing the bound ozone. This is a speculative example of what could lead to a reduction in deposition rates.

Overall the two-layer mechanism provides a good representation of the temperature and wind speed dependence and absolute values of ozone deposition to the ocean. It is also sufficiently simple mathematically to require relatively little extra
time to compute within the framework of the ACCESS chemistry-climate model.

## 7 Global fields of ozone deposition velocity, loss and concentration

### 7.1 Deposition velocity

The modelled global distribution of ozone deposition velocity (cm s$^{-1}$) averaged over the year 2006 in Figure 9 shows that the two-layer reactivity scheme (top panel) gives the largest deposition velocities, as high as 0.03 cm s$^{-1}$, in the tropics. $v_d$
decreases with increasing latitude, but at around 60°S it increases again slightly mainly due to the increase in ozone solubility with decreasing water temperature starts to exert more influence coupled with higher wind speeds in this latitudinal zone. From the tropics to mid latitudes the global distribution of $v_d$ is qualitatively similar to the global SST distribution (plot not shown) because of the dominance of the ozone-iodide reaction in determining $v_d$ in which iodide concentration has a proportional dependence on SST. The distribution from the one-layer reactivity scheme (middle panel) is
qualitatively similar to the two-layer result but with $v_d$ values almost 2 to 3 times as large. The default ACCESS-UKCA scheme with $r_c = 2200$ s m$^{-1}$ (bottom panel) provides a much flatter variation of $v_d$ across the globe. The magnitude of deposition velocity predicted by this scheme within the tropics is similar to that by the one-layer scheme, but beyond the tropics the values predicted by the constant surface resistance model are larger by approximately a factor of two.

The relative difference (%) between $v_d$ predicted by the one-layer ($v_{d1}$) and two-layer ($v_{d2}$) reactivity schemes (defined as
$(v_{d1} - v_{d2}) \times 100 / v_{d2}$) is presented in Figure 10 (top panel), where differences as high as 150% can be seen in the mid latitudes, and as low as 25–50% in the tropics. The differences between the default scheme and the two-layer reactivity scheme (bottom panel) in the mid latitudes are even greater (up to 300%).

In Table 1, the average deposition velocity for seawater (which excludes sea ice and coastal grid cells) obtained using the two-layer reactivity scheme is almost half that of the default scheme and two-third that of the one-layer scheme, which leads
to a small lowering of the globally averaged value for the two-layer scheme. The averaged Southern Hemisphere deposition velocity values are generally lower than the Northern Hemisphere values.

To explore seasonal differences, we plot Figure 11 which is the same as Figure 9 except that the left panels are for January 2006 and the right panels are for July 2006. The distributions are similar to the corresponding ones in Figure 9 but there is more spatial variability in Figure 11 because of shorter averaging (i.e. one month). For the two-layer scheme (top panels), in





January the deposition velocities are generally larger in the mid to high latitudes in the Northern Hemisphere than those in the Southern Hemisphere, and as expected this behaviour is reversed in July mostly as a result of seasonal SST changes. For such latitudes in the Northern Hemisphere, the January deposition velocities are higher than those in July, and the reverse occurs in the Southern Hemisphere. In the tropics and subtropics, there is an upward latitudinal shift of high deposition velocity regions in July compared to January. In qualitative terms, the above behaviour is also evident in the results from the one-layer scheme (middle panels). For the default scheme (bottom panels), the seasonal variations in $v_d$ are smaller and less apparent, and would be mainly dominated by seasonal wind-speed variations. In all plots, the white region around the poles is the extent of sea ice which is larger in July around the South Pole and in January around the North Pole.

Ganzeveld et al. (2009) presented January and July deposition velocities simulated using the one-layer reactivity scheme in their free running ECHAM5/MESSy climate-chemistry model (see their Figures 3a and 3b, respectively). Their plots are significantly different to the corresponding middle panels in Figure 11. Our results show maximum values of deposition velocity in the tropics and subtropics (largely as a result of enhancement due to chemical reaction because of higher iodide associated with higher SSTs) whereas the maxima in their plots are mostly located in mid-latitudes which they attribute to high wind speeds in the storm track regions located at such latitudes. Apart from potentially significant differences in the formulation of the two climate-chemistry models, there are other differences as well such as resolution, use of nudging in our model, emissions, and the parameterisation of iodide concentration.

## 7.2 Ozone loss

The modelled global distribution of dry deposition velocity to the ocean can be combined with the modelled ozone concentration fields to enable the loss of ozone to the ocean surface to be estimated. The deposition budgets obtained using ACCESS-UKCA are presented in Table 2. Compared to the default scheme the global ozone deposition to seawater, excluding sea ice and coastal grid cells, is 85% for the one-layer reactivity scheme and it is almost halved for the two-layer reactivity scheme. Deposition to the sea is 12% of the total deposition for the two-layer scheme, 19% for the one-layer scheme and 21% for the default scheme. The reduction in oceanic deposition using the two-layer scheme corresponds to a 10% decrease in the total global estimate of ozone deposition relative to the original (default) scheme. The seawater deposition in the Northern Hemisphere is slightly larger than that in the Southern Hemisphere due to the higher average seawater deposition velocity in the former (Table 1). Of the total global ozone deposition, about 70% is in the Northern Hemisphere and 30% in the Southern Hemisphere, the former being larger due to the much larger terrestrial contribution. The total ozone deposition obtained from ACCESS-UKCA is on the lower end of the values reported from other models: $1094 \pm 264$ Tg yr$^{-1}$ (Young et al., 2013), $1003 \pm 200$ Tg yr$^{-1}$ (Stevenson et al., 2006) and $949 \pm 222$ Tg yr$^{-1}$ (Wild, 2007) (see also Climate Change, The Physical Science Basis, 2013, IPCC, AR5, Chapter 8, Anthropogenic and Natural Radiative Forcing, Table 8.1, page 672; Young et al., 2013). This is likely due to a general underestimation of tropospheric ozone by ACCESS-UKCA (Woodhouse et al., 2015).



### 7.3 Ozone concentration

While we have previously noted that ACCESS-UKCA underestimates tropospheric ozone, it is worthwhile to examine how an oceanic dry deposition parameterisation affects tropospheric ozone distributions. The top panel in Figure 12 shows the modelled annual, near-surface (at 20 m) ozone concentration (ppbv) based on the two-layer reactivity scheme. Relatively

high concentrations in the Northern Hemisphere, especially within 0–50°N, are evident, which can be related to the larger precursor emissions in these regions. The middle panel presents the relative difference (%) between the ozone concentration predicted by the one-layer ($c_1$) and two-layer ($c_2$) reactivity schemes (defined as $(c_1 - c_2) \times 100 / c_2$). It can be seen that the largest underestimation of the near surface ozone concentration by the one-layer scheme is 11% for 45–70°S. The bottom panel in Figure 12 is the relative difference between the concentration predicted by the default scheme and the two-layer

reactivity scheme, which shows the largest underestimation by the default scheme of 17% within the same latitudinal band. The bottom panel can be approximately compared with the results in Figure 10 of the paper by Ganzeveld et al. (2009) in which they present the relative difference between a constant $r_c$ scheme (with $r_c = 2000$ s m$^{-1}$) and the one-layer scheme. The largest underestimation by their constant $r_c$ scheme is about 4% in the latitudes 45–70°S; this latitudinal band is very similar to what is suggested by the present results but with an underestimation of 17%. Their results show that there is also a

significant overestimation by their constant $r_c$ scheme, the largest of which is about 4% over the tropical and subtropical (30°S–30°N) waters and in high latitude regions in the Northern Hemisphere. This, however, is not apparent in our results.

We also examine the influence of the various schemes on the modelled ozone distribution throughout the troposphere. The top left plot in Figure 13 is the observed zonal distribution of ozone concentration as a function of altitude for the year 2006 based on the monthly mean ozone profile database available from and described at

http://www.bodekerscientific.com/data/monthly-mean-global-vertically-resolved-ozone. These gap filled data (Tier 1.4) are based on the raw individual ozone data sourced from the so-called Binary Data Base of Profiles (BDBP) database (Hassler et al., 2009) and cover the whole globe. The top right plot is the corresponding distribution obtained from ACCESS-UKCA using the two-layer reactivity scheme. There are similarities between the observed and modelled fields; for example, the tropospheric ozone levels are lower in the Southern Hemisphere in both plots. The high ozone concentration areas at the top

corners correspond to the lower edges of the stratosphere which starts at an altitude of about 10 km at the poles and increase to about 15 km at the Equator. In the troposphere, the modelled concentrations are generally lower compared to the observations, for example in the high latitudes, which is a known feature of the model (Woodhouse et al., 2015). This is more clearly evident in the bottom-left plot which presents the relative difference (= $[(\overline{M} - \overline{O}) / \overline{O}] \times 100$ %) between the time-averaged modelled concentration ($\overline{M}$) determined using the two-layer reactivity scheme and the time-averaged

observed concentration ($\overline{O}$). The bottom-right figure is the same plot except for the default scheme. There are improvements in the ozone prediction with the two-layer scheme south of 40°S, both at the lowest altitudes and at upper





levels as high as 6 km. There is also some improvement in the tropical south at a height of about 8 km. There is a slight model improvement in the high latitudes 70–80°N between the altitudes 2–6 km. Overall, for the latitudes 45–70°S, the two-layer scheme improves the prediction of the observed zonal ozone mixing ratios at an altitude of 1 km by 7% compared to the original default scheme. For the altitude range 1–6 km, this improvement is 5%.

**8 Conclusions**

Ozone dry deposition parameterisation schemes are best evaluated within an integrated modelling framework that includes multi-parameter interdependencies of deposition as exists in the field. Using recent measurements, we assessed the performance of three ozone dry deposition schemes for seawater within the global climate-chemistry model ACCESS-UKCA incorporating meteorological nudging and monthly-varying emissions. The default schemes  assumes a constant

water surface resistance of 2200 s m$^{-1}$, which is a commonly used assumption in most atmospheric chemistry/composition models. The second scheme is a mechanistic, one-layer reactivity scheme proposed by Fairall et al. (2007) in which the surface resistance formulation includes the simultaneous influence of ozone solubility in water, waterside molecular diffusion and turbulent transfer, and a first-order chemical reaction of ozone with dissolved iodide. Third scheme is a development of Fairall et al.'s (2007) two-layer reactivity scheme in which, unlike the single, uniform reactivity assumption

in the one-layer scheme, the water surface has a high reactivity and the region below has a very small background reactivity.

A comparison of the observed deposition velocity dependencies on sea-surface temperature and wind speed with those obtained from ACCESS-UKCA using the three schemes showed that the two-layer scheme is able to describe the absolute magnitude and the sea surface temperature and wind speed dependence of the field measurements most closely. The two-layer scheme results are sensitive to the value of the thickness of the surface reactive layer – a value between 2 to 3 microns,

which is about the same as the typical react-diffusive length scale over which the ozone-iodide chemical reaction dominates deposition, works well and we chose 2.5 microns. The assumption in the first scheme of a constant water surface resistance overestimates the observed deposition velocity by a factor of 2 to 4 and does not describe its variability with SST and wind speed. The one-layer scheme performs somewhat better than the default scheme. Although the one-layer scheme includes all the important processes as the two-layer scheme, it overestimates the influence of waterside turbulent transfer on chemical

reaction when the reaction is slow (which occurs under colder SSTs in the Southern Ocean), which results in an overestimate of the observed deposition velocity by a factor of 2 to 3. The two-layer scheme indirectly limits this influence beyond the reactive layer.

The modelled global distributions show that the two-layer reactivity scheme yields maximum ozone deposition velocities in the tropics, they decrease with increasing latitude and then increase again slightly in the Southern Hemisphere. Significant

local variations are also evident. This latitudinal behaviour is the result of the combined effects of chemical reaction,





solubility, molecular diffusion and turbulent transfer, and because all these processes except turbulent transfer are direct functions of SST, which in turn is a function of latitude, there is a strong latitudinal dependence.

Previous model estimates in the scientific literature suggest approximately one third of the total ozone dry deposition is to the ocean. The deposition velocities obtained using the two-layer scheme and the recent observations point to a substantially lower ozone deposition to the ocean. A new modelled estimate of ozone deposition to the ocean of about 80 Tg yr$^{-1}$ was obtained (which excludes sea ice and coastal grid cells), which is 12% of the modelled total global ozone deposition, is almost half of the original oceanic deposition obtained using the default scheme, and corresponds to a 10% decrease in the total global estimate of ozone deposition relative to the original (default) scheme, making a significant change to the overall tropospheric ozone budget.

The effects of the choice of a deposition scheme in the global model were noticeable in the predicted ozone mixing ratios, especially in the Southern Hemisphere. For the latitudes 45–70°S, the two-layer scheme improved the ozone predictions near the surface by 7% and those within the altitude range 1–6 km by 5% compared to the original default scheme.

Based on all the evaluation results presented above, the two-layer reactivity scheme as formulated in this paper is the best performing scheme for describing the ozone deposition to seawater in a global modelling framework.

Further observations on deposition velocity with greater temporal and spatial coverage would help constrain dry deposition schemes better. Studies involving observations of the sea surface microlayer, the role of wave breaking and bubbles on deposition processes, and the degree of relevant feedbacks in climate-chemistry models (e.g., the ozone reaction with oceanic iodide which produces iodine compounds that participate in atmospheric ozone destruction cycles in the troposphere) need to be explored.

**Acknowledgements**

The work was supported by funding from the Australian Climate Change Science Program (ACCSP). The model runs were performed on the National Computational Infrastructure (NCI) Facility in Canberra, Australia, which is supported by the Australian Commonwealth Government. Communication with Chris Fairall, Detlev Helmig, Laurens Ganzeveld and Rosie Chance on their works was very useful. We are indebted to Ludovic Bariteau for kindly supplying data related to their published work. Fiona O'Connor, Mohit Dalvi, Steve Rumbold and Colin Johnson of the U.K. Met Office are thanked for their assistance with the UKCA emission methodology and answering questions about UM-UKCA. Luke Abraham of University of Cambridge is thanked for his help with several technical questions on UKCA. Martin Dix and Scott Wales are acknowledged for their help with model configuration issues, and Lauren Stevens for help with emissions data. ERA-Interim data from the European Centre for Medium-Range Weather Forecasts was used in this research. We would like to thank Greg Bodeker (Bodeker Scientific) and Birgit Hassler (NOAA) for providing the combined vertical ozone profile database.



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



**Table 1: Modelled average ozone dry deposition velocities for the year 2006 (cm s⁻¹)**

| Deposition scheme | Ocean | | | Global | | |
|---|---|---|---|---|---|---|
| | Northern Hemisphere | Southern Hemisphere | Average | Northern Hemisphere | Southern Hemisphere | Average |
| Two layer | 0.020 | 0.017 | 0.018 | 0.097 | 0.051 | 0.074 |
| One layer | 0.033 | 0.032 | 0.032 | 0.104 | 0.062 | 0.083 |
| Default (constant $r_c$) | 0.039 | 0.039 | 0.039 | 0.108 | 0.067 | 0.087 |

**Table 2: Modelled ozone dry deposition for the year 2006 (Tg yr⁻¹)**

| Deposition scheme | Ocean | | | Global | | |
|---|---|---|---|---|---|---|
| | Northern Hemisphere | Southern Hemisphere | Total | Northern Hemisphere | Southern Hemisphere | Total |
| Two layer | 44.8 | 37.7 | 82.5 | 500.5 | 209.2 | 709.7 |
| One layer | 76.5 | 67.5 | 144.0 | 530.4 | 237.7 | 768.1 |
| Default (constant $r_c$) | 89.2 | 78.5 | 167.7 | 541.5 | 248.0 | 789.5 |



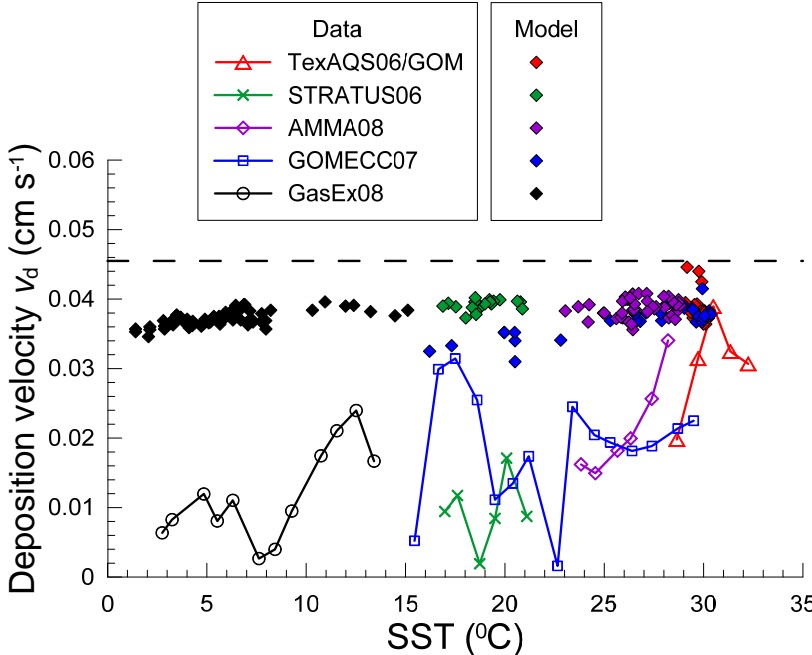

**Figure 1: Ozone dry deposition velocity ($v_d$) as a function of sea surface temperature (SST) from five field experiments (Helmig et al., 2012; Ludovic Bariteau, personal communication, 2016) and the corresponding values obtained from the ACCESS-UKCA model using the default parameterisation for ozone deposition to the ocean involving a constant $r_c = 2200$ s m⁻¹.**





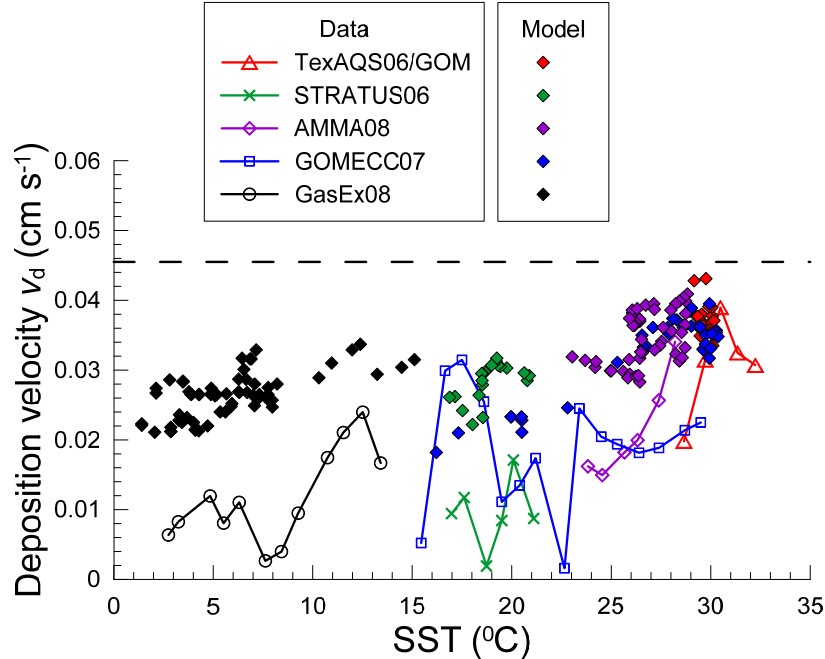

**Figure 2: Ozone dry deposition velocity ($v_d$) as a function of sea surface temperature (SST) from five field experiments (Helmig et al., 2012; Ludovic Bariteau, personal communication, 2016) and the corresponding values obtained from the ACCESS-UKCA model using the one-layer scheme for ozone deposition to the ocean.**



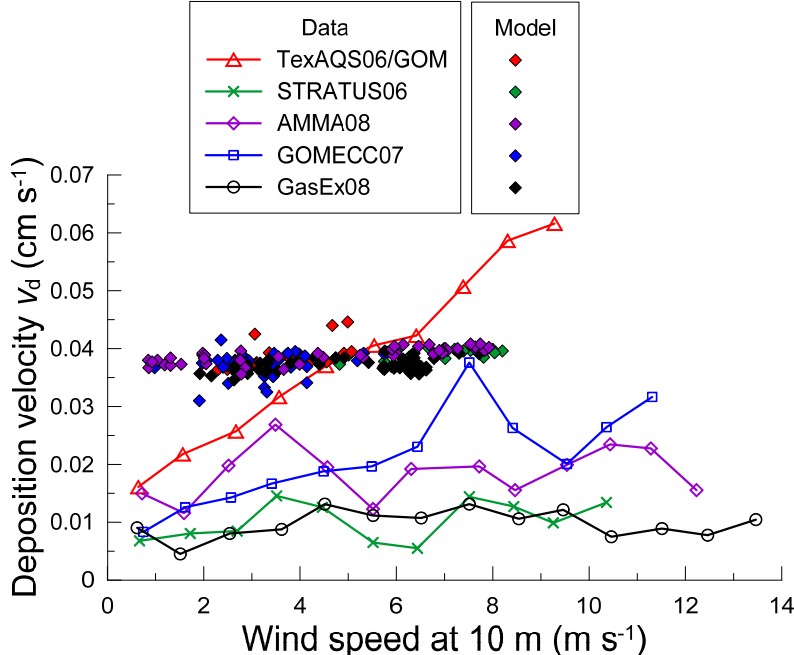

**Figure 3: Ozone dry deposition velocity ($v_d$) as a function of wind speed at 10 m height from five field experiments (Helmig et al., 2012) and the corresponding values obtained from the ACCESS-UKCA model with the default parameterisation for ozone deposition to the ocean.**





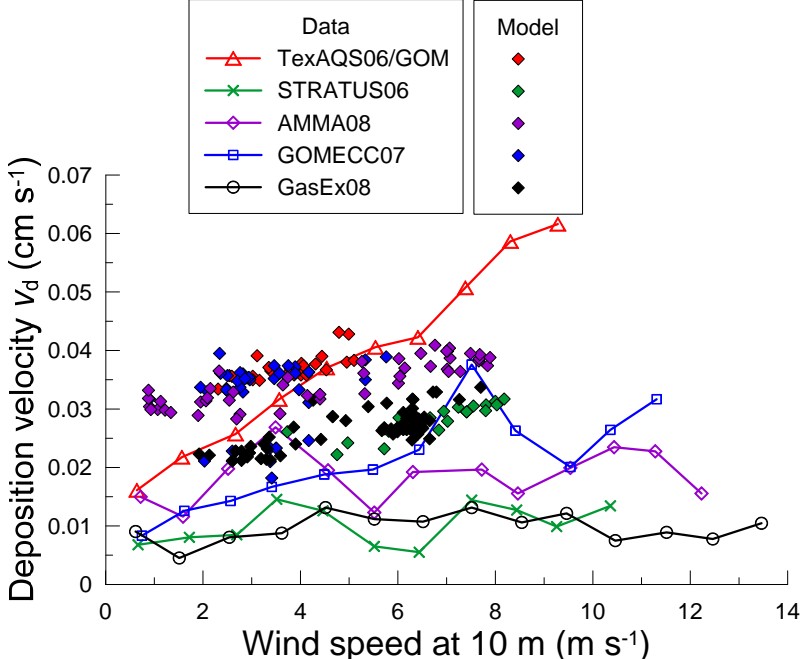

**Figure 4: Ozone dry deposition velocity ($v_d$) as a function of wind speed at 10 m height from five field experiments (Helmig et al., 2012) and the corresponding values obtained from the ACCESS-UKCA model using the one-layer scheme for ozone deposition to the ocean.**





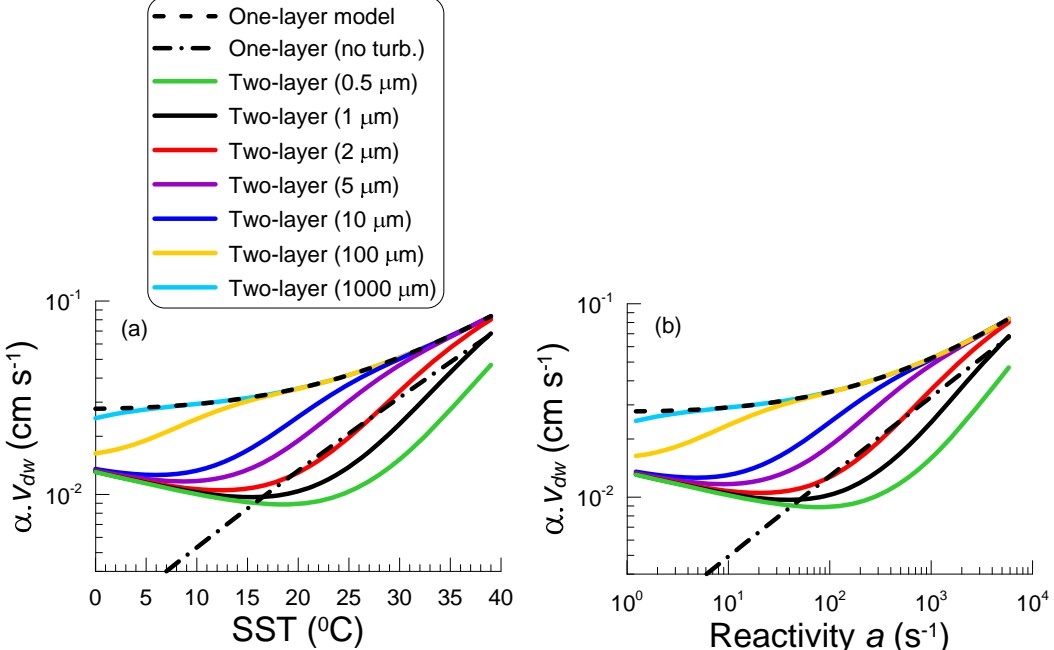

**Figure 5: Variation of the oceanic component of ozone dry deposition velocity** $\alpha v_{dw}$ $(=1/r_c)$ **as a function of (a) sea surface temperature (SST, °C), and (b) reactivity** $a$ **(s$^{-1}$), determined using the one-layer deposition scheme with and without waterside turbulence, and the two-layer deposition scheme for a range of** $\delta_n$ **values. Eq. (17) was used for the iodide concentration and the waterside friction velocity** ($u_{*_w}$) **used was 0.01 m s$^{-1}$.**





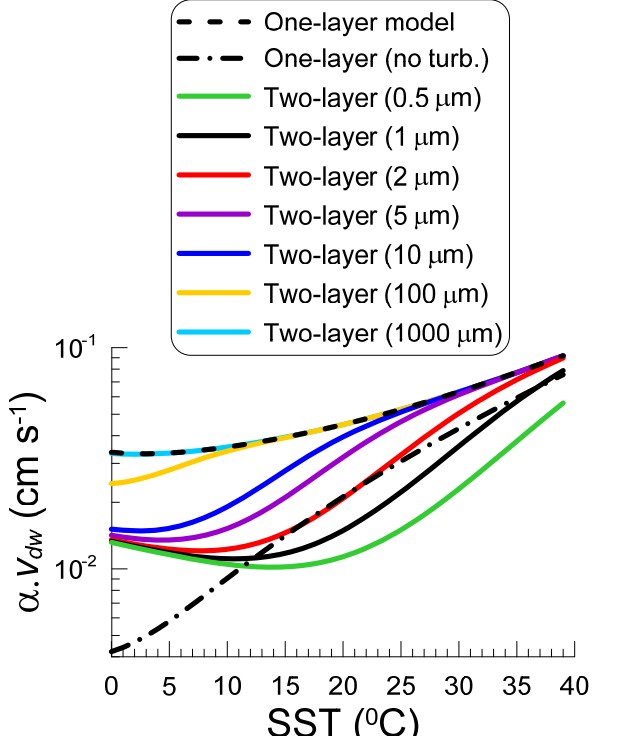

Figure 6: Variation of the oceanic component of ozone dry deposition velocity $\alpha v_{dw}$ ($= 1/r_c$) as a function of sea surface temperature (SST, °C) determined using the one-layer deposition scheme with and without waterside turbulence, and the two-layer deposition scheme for a range of $\delta_n$ values. Eq. (24) was used for the iodide concentration and the waterside friction velocity ($u_{*_w}$) used was 0.01 m s⁻¹.





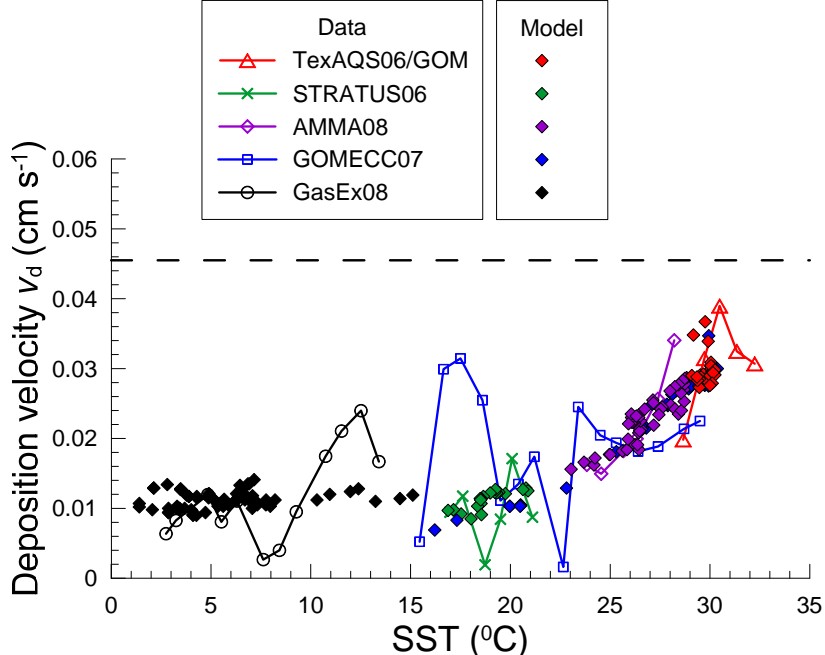

**Figure 7: Ozone dry deposition velocity ($v_d$) as a function of sea surface temperature (SST) from five field experiments (Helmig et al., 2012; Ludovic Bariteau, personal communication, 2016) and the corresponding values obtained from the ACCESS-UKCA model using the two-layer reactivity scheme for ozone deposition to the ocean.**





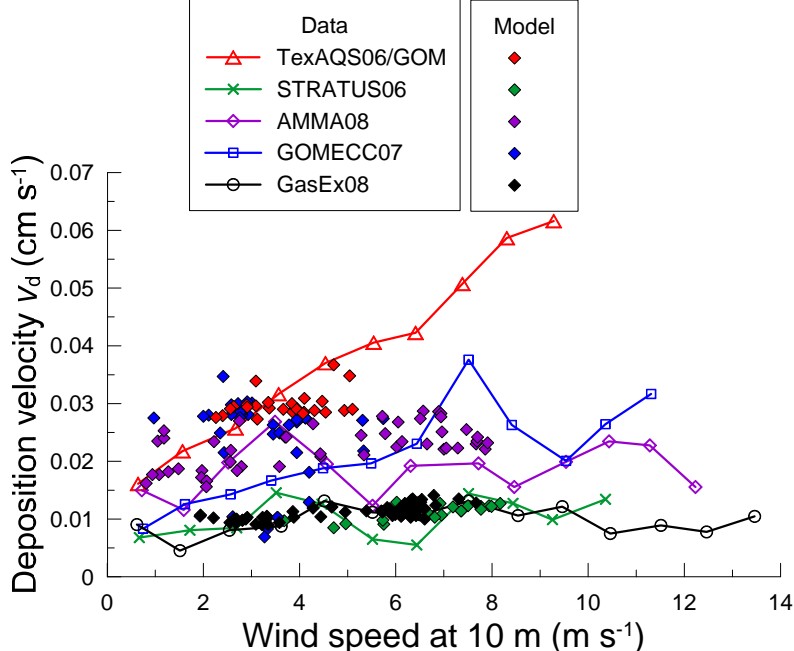

**Figure 8: Ozone dry deposition velocity ($v_d$) as a function of wind speed at 10 m height from five field experiments (Helmig et al., 2012) and the corresponding values obtained from the ACCESS-UKCA model using the two-layer reactivity scheme for ozone deposition to the ocean.**



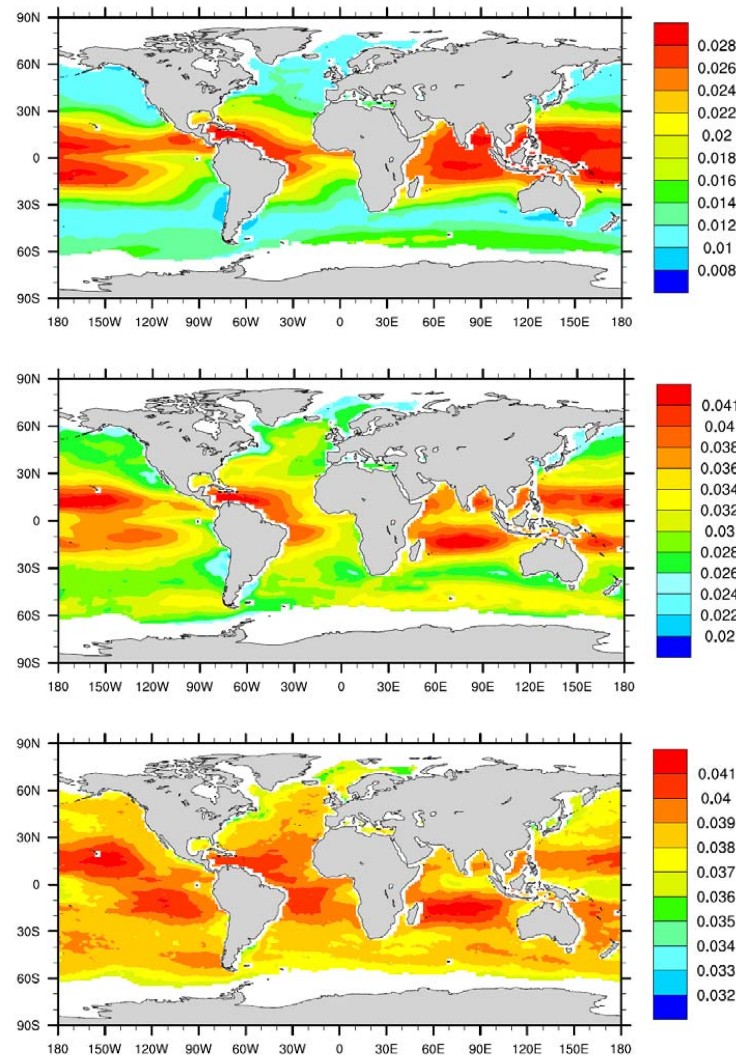

**Figure 9: Modelled ozone dry deposition velocity ($v_d$, cm s$^{-1}$) averaged over the year 2006 obtained using the two-layer reactivity scheme (top panel), the one-layer reactivity scheme (middle panel), and the default scheme with $r_c = 2200$ s m$^{-1}$ (bottom panel). Note differences in scale.**





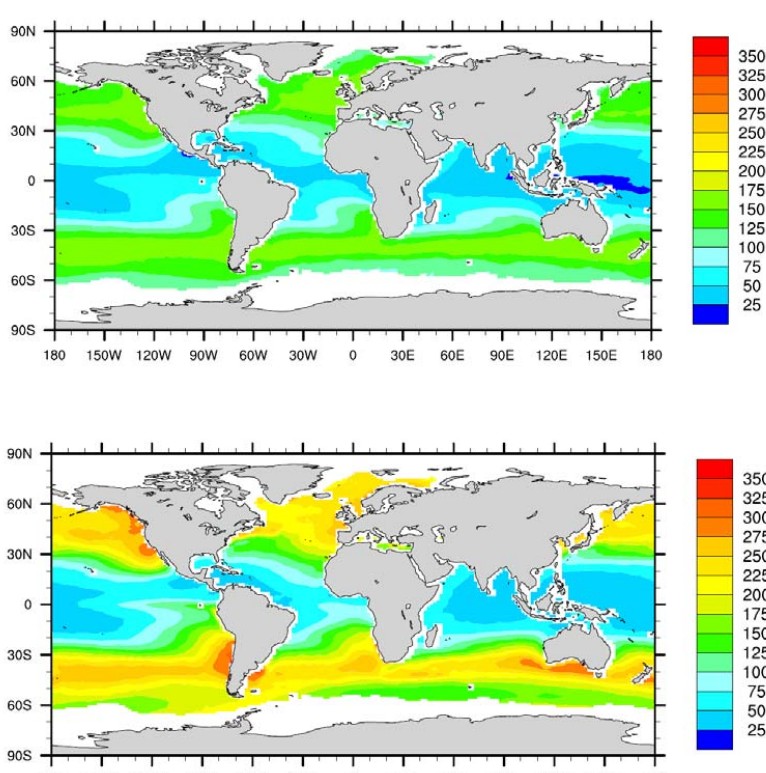

5    **Figure 10: Average relative difference (%) for the year 2006 between the ozone dry deposition velocity ($v_d$, cm s$^{-1}$) modelled using the one-layer reactivity scheme and the two-layer reactivity scheme (top panel), and that between the default scheme (with $r_c = 2200$ s m$^{-1}$) and the two-layer reactivity scheme (bottom panel).**





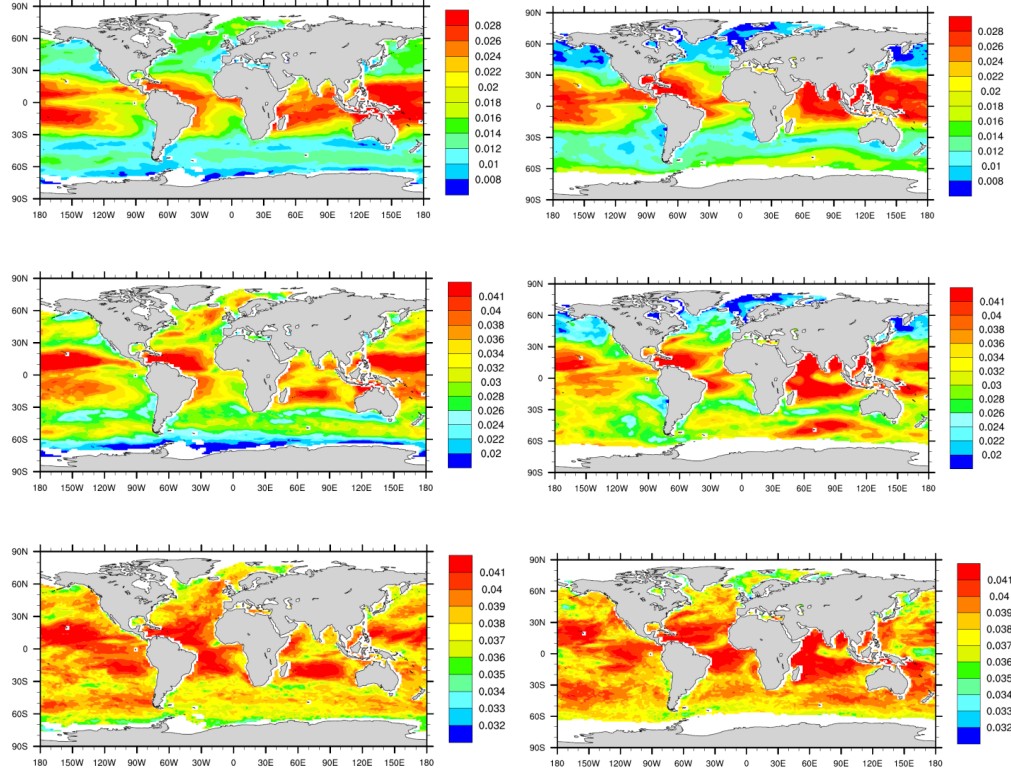

5  **Figure 11: Modelled ozone dry deposition velocity ($v_d$, cm s$^{-1}$). The left panels are for January 2006 and right panels are for July 2006. Top panels: the two-layer reactivity scheme, middle panels: the one-layer reactivity scheme, and bottom panels: the default scheme with $r_c = 2200$ s m$^{-1}$. Note differences in scale.**





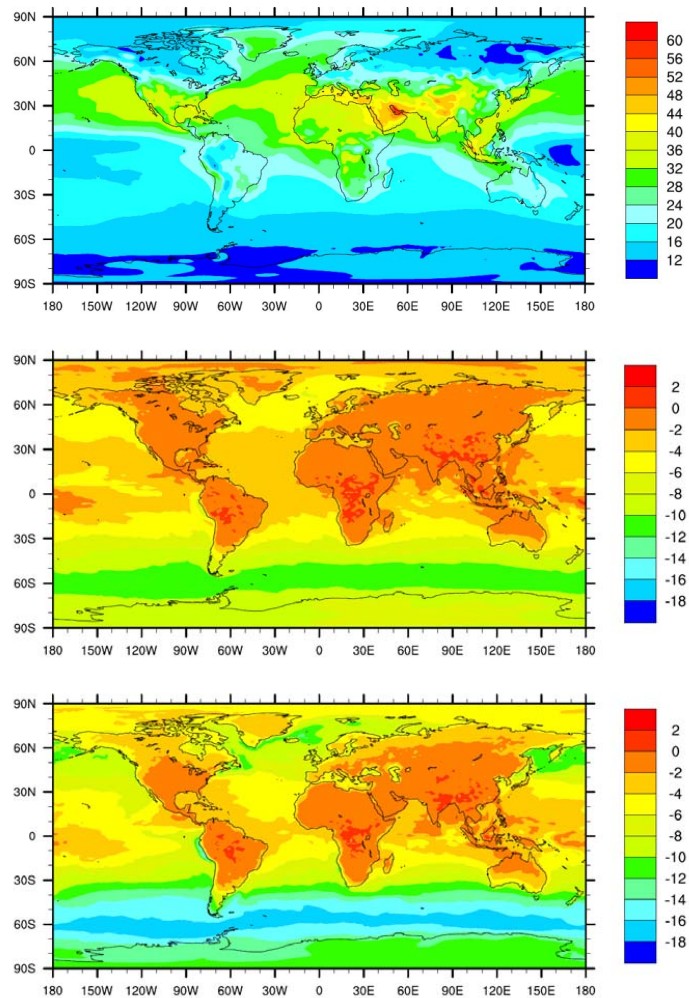

5    **Figure 12: Top panel: Modelled annual ozone concentration (ppbv) for 2006 obtained using the two-layer reactivity scheme; middle panel: average relative difference (%) between the ozone concentration predicted using the one-layer reactivity scheme and the two-layer reactivity scheme; and bottom panel: average relative difference (%) between the ozone concentration predicted using the default scheme with $r_c = 2200$ s m$^{-1}$ and the two-layer reactivity scheme.**



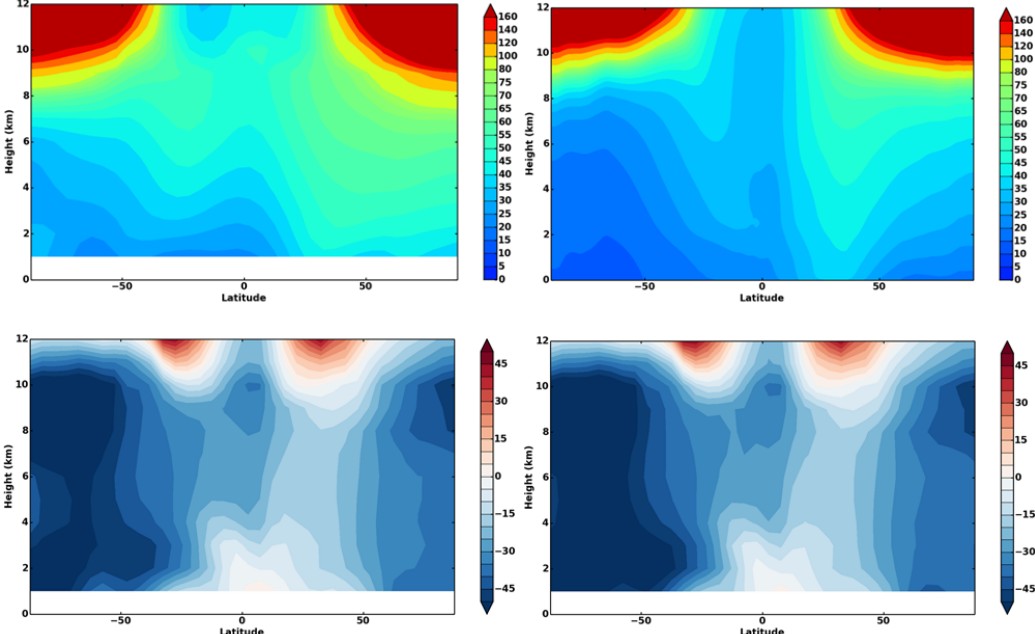

**Figure 13: Zonal distribution of ozone concentration (ppbv) for the year 2006. Top left: observed distribution based on the global monthly mean vertical ozone profile database available from http://www.bodekerscientific.com; top right: the corresponding ACCESS-UKCA model distribution using the two-layer reactivity scheme; bottom left: the relative difference (%) between the modelled concentration determined using the two-layer reactivity scheme and the observations; and bottom right: the relative difference (%) between the modelled concentration determined using the default scheme and the observations.**