# Peer review of "An improved parameterisation of ozone dry deposition to the ocean and its impact in a global climate-chemistry model"

_Atmospheric Chemistry and Physics, 2016_

## Referee Comment (RC1) · C. Fairall (Referee) · 10 Nov 2016

This paper describes an improved parameterization for oceanic deposition of ozone. It is based on a 2-layer molecular/turbulence model where the reactive component on the Oceanside is restricted in depth. The concentration of the reactive component (iodide) is represented as a function of ocean temperature. The new parameterization is compared to a set of direct observations from shipboard measurements. It gives a much better fit to the data compared to the 1-layer approach. The parameterization is incorporated into a global chemistry model and the results with different parameterizations are compared. The basic approach is sound and the new parameterization fits the data very well.

In my view the paper is acceptable for publication in its present form. One essential point is the restriction of the reaction to within 2 microns of the surface. This seems artificial and, as discussed by the authors on page 13, may be a surrogate for the decrease of turbulence near the surface because of dissipation. Perhaps this issue could be solved with a better representation of turbulent mixing, near the interface, but for now their method is successful as a parameterization that seems to work. The authors may wish to reiterate this point in their conclusions.

Here are a few other comments: *Figure 5b is confusing in that it appears that an increase in a can lead to a decrease in Vdw. I finally figured out that reactivity is not an independent variable but correlated with temperature. *Figure 7. It would be amusing to see the 1-layer no turbulence solution on this graph.

---

## Referee Comment (RC2) · Anonymous Referee #2 · 22 Nov 2016

**Overview:**

This paper reports the implementation of 2 additional schemes for ozone deposition over oceans into a global chemistry-climate model (the Australian Community Climate and Earth-System Simulator). Model outputs using the original and the 2 additional schemes are compared to recent observations of ozone deposition fluxes and velocities over sea water made during a number of ship cruises. Finally, the impact of the different ozone deposition schemes on atmospheric ozone concentrations is investigated.

Deposition of ozone to the Earth's surface is an important component of the tropospheric $O_3$ budget. The analysis of the HTAP multi-model outputs reported in the cited

paper by Hardacre et al. is indicative of the uncertainty in deposition to both the land and ocean surfaces. The larger surface area of the oceans more than compensates for the smaller ozone deposition velocities to ocean surfaces compared to land, especially vegetation, surfaces. As also shown in Hardacre et al., the deposition to the ocean has the largest uncertainty.

The two additional schemes (denoted 'one-layer' and 'two-layer') both improve the model performance against observations. The 'two-layer' scheme gives the best realisation of the observed deposition velocity and its dependence on sea surface temperature and wind speed (for the assumptions made and parameters used).

The paper addresses a neglected but important topic and covers a relevant subject for this journal. Overall, the paper reads well and should be published, after addressing the specific and technical comments below. These are mostly minor or suggestions to improve clarity.

**Specific Comments:**
Deposition Schemes: The 'one-layer' scheme is effectively that developed by Fairall et al. (2007). The 'two-layer' scheme includes a surface microlayer layer and some pragmatic choices are made to make it computationally tractable. The description of the schemes includes a significant number of equations, which for the one-layer scheme repeat to some extent those given in Fairall et al. (2007). As a suggestion, the authors may wish to consider moving these to an appendix or supplement to aid clarity.

Iodide fields: Based on earlier experimental studies, the $O_3$ deposition process includes *inter alia* a reaction of $O_3$ with iodide ($I^-$) in the surface layers of the ocean (page 7, line15). Initially, I could not see the relevance of the subsequent chemistry (equations 6 to 10 on page 8), except perhaps to indicate that the iodide is effectively regenerated (or how $I_2$ is formed). Are these equations needed?

The cited paper by Ganzeveld et al. (2009) used sea-water nitrate as a proxy to derive global iodide fields. In this study, sea-surface temperatures are used (after Sherwen et al.). Later in the paper (page 18, line 9 and Figure 11), a qualitative comparison is made with the modelled $O_3$ deposition velocity maps in Ganzeveld et al. (2009). There were latitudinal differences in the position of the maxima. One can arguably see the same mid-latitude Atlantic feature that Ganzeveld et al. reported (for July 2006). Allowing for the other differences mentioned (page 18, line 15), it would have been interesting to see a simulation using an iodide field based on nitrate.

Nudging and monthly-averaged fields: The model is nudged to ERA-Interim reanalyses of wind and temperature. Thus, synoptic variability is similar to that observed, improving the comparison with observations of atmospheric trace constituents.

Monthly-averaged concentration fields are used in the analysis reported here. To those more familiar with land-based deposition, the use of monthly-averaged fields would not seem realistic, given the strong diurnal variations in $O_3$ concentrations and in the stomatal uptake over vegetation-cover surfaces. It might be worth contrasting this with the behaviour over oceans where the diurnal variations are expected to be smaller (presumably), which justifies the use of monthly-averaged fields. What about coastal regions, which are more influenced by the land???

The UKCA atmospheric chemistry module used in ACCESS has a branch, which includes an aircraft/satellite emulator, allowing the model to be sampled at the time and location of an aircraft or satellite sounding (http://www.ukca.ac.uk/wiki/index.php/Flight_track_emulator_to_compare_to_campaign_data). It could easily be used for ship-based measurements. The model could then be sampled at the actual time of the observation.

Performance of standard model configuration: The modelled $O_3$ concentrations and budgets are underpredicted. While this does not detract from the present work, it

would be good to understand the source of this bias. The cited paper by Woodhouse et al. does not really provide further details as to whether this is a result of emissions, initialisation, spin-up, etc ...

Code availability Are these model developments available in an accessible code repository? If so, the authors could provide access details as I am sure that other groups would be interested in using this approach.

**Technical comments:**

It was hard to compare Figures 5a and 6, especially on the screen. As a suggestion, I wonder if these could be combined into a single plot or made into a single figure on the same page to make the comparison easier.

The colour scale used in the panels of Figures 9 and also of Figure 11 differ. While the colour scales used aid the comparison of the spatial patterns, it makes comparison of the the absolute magnitudes harder. It might be better if a common scale could be adopted, with sufficient dynamic range and discretisation.

---

## Author Comment (AC1) · 15 Dec 2016

We thank Dr. Fairall for reviewing the manuscript and providing very useful comments. In the following, we provide a response to the comments made by him (the Referees' comments are given in inverted commas).

Comment: "This paper describes an improved parameterization for oceanic deposition of ozone. It is based on a 2-layer molecular/turbulence model where the reactive component on the Oceanside is restricted in depth. The concentration of the reactive component (iodide) is represented as a function of ocean temperature. The new parameterization is compared to a set of direct observations from shipboard measurements. It gives a much better fit to the data compared to the 1-layer approach. The

parameterization is incorporated into a global chemistry model and the results with different parameterizations are compared. The basic approach is sound and the new parameterization fits the data very well."

Response: Thank you for your comment.

Comment: "In my view the paper is acceptable for publication in its present form. One essential point is the restriction of the reaction to within 2 microns of the surface. This seems artificial and, as discussed by the authors on page 13, may be a surrogate for the decrease of turbulence near the surface because of dissipation. Perhaps this issue could be solved with a better representation of turbulent mixing, near the interface, but for now their method is successful as a parameterization that seems to work. The authors may wish to reiterate this point in their conclusions."

Response: The ozone-iodide reaction in seawater is fast such that the bulk of it takes place within the thin viscous sublayer near the water surface. In the current parameterisation, the waterside eddy diffusivity is assumed to vary linearly with depth. This assumption is valid for a fully turbulent flow and overestimates turbulent transfer in the viscous sublayer because it does not account for viscous dissipation of turbulence in this layer. This overestimation leads to a stronger chemistry-turbulence interaction, with the overall impact being an overestimation of ozone deposition velocity using the one-layer reactivity scheme.

As mentioned by the referee and discussed on pages 13 and 16, the two-layer reactivity scheme limits the iodide concentration to a specified depth from the water surface, thereby restricting the ozone-iodide reaction to that depth. Although this restriction is artificial, it works as surrogate to compensate for the overestimation of turbulent mixing in the viscous sublayer. A better parameterisation of turbulent mixing near the interface in the concentration conservation equation could overcome the issue. However, as commented by the referee, for now the two-layer method is successful and is adopted as a parameterization for deposition velocity. We will reiterate this point in the conclusions.

Comment: "Here are a few other comments: *Figure 5b is confusing in that it appears that an increase in a can lead to a decrease in Vdw. I finally figured out that reactivity is not an independent variable but correlated with temperature. *Figure 7. It would be amusing to see the 1-layer no turbulence solution on this graph."

Response: Yes, that is correct. The y-axis in Fig. 5b is 'alpha*V_dw'. Reactivity (a) is the product of the second-order rate coefficient and iodide concentration (Eq. 5), both of which increase with sea surface temperature (SST) and result in an increase of V_dw with SST. However, the ozone solubility in water (alpha) increases with decreasing SST. As mentioned in the first paragraph on Page 15, "there is a slight increase in 'alpha*V_dw' with decreasing reactivity, which is mostly due to the larger influence of solubility which increases with decreasing temperature."

Regarding "Figure 7. It would be amusing to see the 1-layer no turbulence solution on this graph," we plot the one-layer no turbulence solution in the following figure (represented by small dots). The bottom figure is the same as the top one except that the points predicted by the two-layer model are not plotted for clarity. At SSTs greater than 17 deg C, the one-layer (no turbulence) and two-layer solutions are very similar as chemical reaction dominates over turbulence. For SSTs less than 17 deg C, the one-layer (no turbulence) solution predicts progressively lower deposition velocities than the two-layer solution because it neglects turbulence whose influence becomes important compared to weaker chemical reaction at such temperatures.

We will replace Figure 7 with the attached figure.

[Figure]

[Figure]

Figure: Ozone dry deposition velocity ($v_d$) as a function of sea surface temperature (SST) from five field experiments (Helmig et al., 2012; Ludovic Bariteau, personal communication, 2016) and the corresponding values obtained from the ACCESS-UKCA model using the two-layer reactivity scheme for ozone deposition to the ocean. The small dots represent the values obtained using the one-layer reactivity scheme without oceanic turbulence solution.

**Fig. 1.** Figure 7 with no turbulence solution plotted

[Figure]

---

## Author Response (AR1)

**Reply by the authors to the Referee #1's comments on acp-2016-844**

"An improved parameterisation of ozone dry deposition to the ocean and its impact in a global climate-chemistry model" by Ashok K. Luhar et al.

**Dr. C. Fairall (Referee #1)**

We thank Dr. Fairall for reviewing the manuscript and providing very useful comments. In the following, we provide our response to the comments made by him.

**Comment:** "This paper describes an improved parameterization for oceanic deposition of ozone. It is based on a 2-layer molecular/turbulence model where the reactive component on the Oceanside is restricted in depth. The concentration of the reactive component (iodide) is represented as a function of ocean temperature. The new parameterization is compared to a set of direct observations from shipboard measurements. It gives a much better fit to the data compared to the 1-layer approach. The parameterization is incorporated into a global chemistry model and the results with different parameterizations are compared. The basic approach is sound and the new parameterization fits the data very well."

**Response:** Thank you for your comment.

**Changes in manuscript:** None.

**Comment:** "In my view the paper is acceptable for publication in its present form. One essential point is the restriction of the reaction to within 2 microns of the surface. This seems artificial and, as discussed by the authors on page 13, may be a surrogate for the decrease of turbulence near the surface because of dissipation. Perhaps this issue could be solved with a better representation of turbulent mixing, near the interface, but for now their method is successful as a parameterization that seems to work. The authors may wish to reiterate this point in their conclusions."

**Response:** The ozone-iodide reaction in seawater is fast such that the bulk of it takes place within the thin viscous sublayer near the water surface. In the current parameterisation, the waterside eddy diffusivity is assumed to vary linearly with depth. This assumption is valid for a fully turbulent flow and overestimates turbulent transfer in the viscous sublayer because it does not account for viscous dissipation of turbulence in this layer. This overestimation leads to a stronger chemistry-turbulence interaction, with the overall impact being an overestimation of ozone deposition velocity using the one-layer reactivity scheme.

As mentioned by the referee and discussed on Pages 13 and 16, the two-layer reactivity scheme limits the iodide concentration to a specified depth from the water surface, thereby restricting the ozone-iodide

reaction to that depth. Although this restriction is artificial, it works as surrogate to compensate for the overestimation of turbulent mixing in the viscous sublayer. A better parameterisation of turbulent mixing near the interface in the concentration conservation equation could overcome the issue. However, as commented by the referee, for now the two-layer method is successful and is adopted as a parameterization for deposition velocity. We have reiterated this point in the 2nd paragraph of Conclusions.

**Changes in manuscript:** The point is reiterated in the 2nd paragraph of Conclusions.

**Comment:** "Here are a few other comments: *Figure 5b is confusing in that it appears that an increase in a can lead to a decrease in Vdw. I finally figured out that reactivity is not an independent variable but correlated with temperature. *Figure 7. It would be amusing to see the 1-layer no turbulence solution on this graph."

**Response:** Yes, that is correct. The y-axis in Fig. 5b is 'alpha*V_dw'. Reactivity (a) is the product of the second-order rate coefficient and iodide concentration (Eq. 5), both of which increase with sea surface temperature (SST) and result in an increase of V_dw with SST. However, the ozone solubility in water (alpha) increases with decreasing SST. As mentioned in the first paragraph of Section 6.1.1 on Page 15, "there is a slight increase in 'alpha*V_dw' with decreasing reactivity, which is mostly due to the larger influence of solubility which increases with decreasing temperature."

Regarding "Figure 7. It would be amusing to see the 1-layer no turbulence solution on this graph," we now plot the one-layer no turbulence solution as Figure 2b (where it is more appropriate than in Figure 7) and describe it in the 2nd last paragraph on Page 12.

**Changes in manuscript:** Figure 2b added and described in the 2nd last paragraph on Page 12 with the text "In Figure 2b we plot the behaviour of the one-layer reactivity scheme with the contribution of waterside turbulence removed. Compared to Figure 2a, the modelled deposition velocities are lower in magnitude and compare better with the data. However, it is clear that for SSTs lower than about 15°C, the model without the waterside turbulence underestimate the deposition velocities compared with the measurements because the influence of waterside turbulence becomes important compared to weaker chemical reaction at lower temperatures. Overall, what this comparison suggests is that the influence of waterside turbulence in the one-layer reactivity scheme is overestimated, and this point will be discussed later on in the paper".

**Reply by the authors to the Referee #2's comments**

"An improved parameterisation of ozone dry deposition to the ocean and its impact in a global climate-chemistry model" by Ashok K. Luhar et al.

**Anonymous Referee #2**

We thank the Referee for reviewing the manuscript and providing very useful comments. In the following, we provide our response to the comments made by the Referee.

**Comment:** "…The paper addresses a neglected but important topic and covers a relevant subject for this journal. Overall, the paper reads well and should be published, after addressing the specific and technical comments below. These are mostly minor or suggestions to improve clarity."

**Response:** Thank you for your comments.

**Changes in manuscript:** None

**Comment:** "Specific Comments:"

"Deposition Schemes: The 'one-layer' scheme is effectively that developed by Fairall et al. (2007). The 'two-layer' scheme includes a surface microlayer layer and some pragmatic choices are made to make it computationally tractable. The description of the schemes includes a significant number of equations, which for the one-layer scheme repeat to some extent those given in Fairall et al. (2007). As a suggestion, the authors may wish to consider moving these to an appendix or supplement to aid clarity."

**Response:** We appreciate the suggestion by the referee, but we think that the number of equations presented in the paper provide a good, self-contained description of the deposition schemes used with logical flow, moving from the simple default scheme to the two-layer reactivity scheme. In doing so, obviously, there is some overlap with relevant equations from the paper by Fairall et al. (2007). We think that keeping the equations in the main text is helpful because they are frequently referred to in the main text when discussing the model intercomparison results and interpreting the model behaviour against the data.

**Changes in manuscript:** None

**Comment:** "Iodide fields: Based on earlier experimental studies, the O3 deposition process includes inter alia a reaction of O3 with iodide (I-) in the surface layers of the ocean (page 7, line15). Initially, I could not see the relevance of the subsequent chemistry (equations 6 to 10 on page 8), except perhaps to indicate that the iodide is effectively regenerated (or how I2 is formed). Are these equations needed?"

**Response:** The referee is correct. The purpose of presenting Equations 6–10 was to highlight the complexity of the ozone-iodide chemistry, to indicate how $I_2$ is generated, and to suggest that there could be a reverse reaction freeing the bound ozone. We have deleted these equations.

**Changes in manuscript:** Equations 6–10 deleted, and modified the relevant sentence in the 2$^{nd}$ last paragraph of Section 6.1.2.

**Comment:** "The cited paper by Ganzeveld et al. (2009) used sea-water nitrate as a proxy to derive global iodide fields. In this study, sea-surface temperatures are used (after Sherwen et al.). Later in the paper (page 18, line 9 and Figure 11), a qualitative comparison is made with the modelled O3 deposition velocity maps in Ganzeveld et al. (2009). There were latitudinal differences in the position of the maxima. One can arguably see the same mid-latitude Atlantic feature that Ganzeveld et al. reported (for July 2006). Allowing for the other differences mentioned (page 18, line 15), it would have been interesting to see a simulation using an iodide field based on nitrate."

**Response:** In our work, the oceanic iodide concentration was parameterised in terms of sea-surface temperature (SST) following MacDonald at al. (2014). A sensitivity analysis was also performed using the parameterisation by Chance at al. (2014) which also used SST (see 2$^{nd}$ paragraph of Section 6.1.1). As mentioned in the 1$^{st}$ paragraph of Page 10, a comprehensive analysis of available iodide measurements conducted by Chance et al. (2014) showed highest iodide concentrations in tropical waters. They examined statistical relationships between iodide and parameters such as SST, nitrate, salinity, chlorophyll-a and mixed layer depth, and found that SST was the strongest predictor of iodide followed by latitude. The correlation ($r^2$) of iodide with $SST^2$ was 0.61 while that with nitrate was 0.36. The parameterisation by MacDonald at al. (2014) is very similar to that by Chance at al. (2014) in that it also involves SST but yields lower iodide magnitudes. Hence the data indicate that the use of SST is preferable to nitrate.

**Changes in manuscript:** None

**Comment:** "Nudging and monthly-averaged fields: The model is nudged to ERA-Interim reanalyses of wind and temperature. Thus, synoptic variability is similar to that observed, improving the comparison with observations of atmospheric trace constituents. Monthly-averaged concentration fields are used in the analysis reported here. To those more familiar with land-based deposition, the use of monthly-averaged fields would not seem realistic, given the strong diurnal variations in O3 concentrations and in the stomatal uptake over vegetation-cover surfaces. It might be worth contrasting this with the behaviour over oceans where the diurnal variations are expected to be smaller (presumably), which justifies the use of monthly-averaged fields. What about coastal regions, which are more influenced by the land???"

**Response:** As stated in the 2$^{nd}$ last paragraph of Page 11 of the paper, oceanic diurnal variations relevant for ozone dry deposition, especially of SST which is a key parameter, are relatively small (also see below the next comment) so the use of monthly-averaged fields can be justified. We have added

appropriate text in the 2$^{nd}$ last paragraph of Page 11 to highlight the fact that there is a strong diurnal variation of land-based deposition in contrast to the weak diurnal dependence of oceanic deposition.

With regards to coastal grid cells, these cover a fraction of the water tile and fractions of other surface tiles. As described in the 2$^{nd}$ last paragraph of Page 6, for a particular grid cell the resistances and the corresponding deposition velocities are calculated for all surface tiles that constitute the overall surface type of that grid cell. A first order loss rate is calculated corresponding to the deposition velocity for each tile, and a grid-cell mean loss rate is obtained by weighting the individual loss rates with the fractions of the surface tile types present in the grid cell. In our analysis for the oceanic deposition, we do not consider any coastal grid cells and only consider those grid cells that are 100% covered by the water tile. We have stated this explicitly in the last paragraph of Page 6.

**Changes in manuscript:** Appropriate text added in the 2$^{nd}$ last paragraph of Page 11 to highlight the fact that there is a strong diurnal variation of land-based deposition. The question about coastal grid is answered in the last paragraph of Page 6.

**Comment:** "The UKCA atmospheric chemistry module used in ACCESS has a branch, which includes an aircraft/satellite emulator, allowing the model to be sampled at the time and location of an aircraft or satellite sounding (http://www.ukca.ac.uk/wiki/index.php/ Flight_track_emulator_to_compare_to_campaign_data). It could easily be used for ship-based measurements. The model could then be sampled at the actual time of the observation."

**Response:** The flight emulator suggested by the referee is part of a UKCA development branch, and not part of any standard release at present. It would require some work on our part to incorporate it into our UKCA branch, and to adapt it to ship based cruises for the present study. However, over the ocean diurnal variations, especially of SST which is shown to be a key parameter for oceanic ozone deposition, are relatively small (which is also mentioned by the Referee above and stated on Page 11 of the paper). Therefore, we do not expect any significant changes in our evaluation results even if an emulator was used. Moreover, we do not have information on the actual timings of the observations along the cruise tracks – only maps of cruise tracks and the number of days over which a cruise was conducted are available. The deposition velocity observations in the paper by Helmig et al. (2012) which we used are only presented as a function of SST and wind speed. We will explore the emulator option for our future model evaluation studies where appropriate.

**Changes in manuscript:** None

**Comment:** "Performance of standard model configuration: The modelled O3 concentrations and budgets are underpredicted. While this does not detract from the present work, it would be good to understand the source of this bias. The cited paper by Woodhouse et al. does not really provide further details as to whether this is a result of emissions, initialisation, spin-up, etc ..."

**Response:** The version of UKCA used in our study underpredicts the observed tropospheric ozone concentration, which is also the reason for the total annual ozone dry deposition obtained from our

model to be on the lower end of the range derived based on the multi-model results given in the paper. We have not been able to pinpoint the reason for the ozone underprediction by UKCA. The ozone bias is an ongoing area of study in ACCESS-UKCA.

**Changes in manuscript:** We say '…the exact reason(s) for which we have not been able to pinpoint but it is an ongoing area of study' at the end of the 1st paragraph of Section 7.2.

**Comment:** "Code availability Are these model developments available in an accessible code repository? If so, the authors could provide access details as I am sure that other groups would be interested in using this approach."

**Response:** We have not yet put these model code changes in any UM-UKCA code repository yet, but intend to do so. The formal process of transferring any model code changes to a UM trunk requires a number of steps. We intend to produce documented code and follow this process up.

**Changes in manuscript:** None

**Comment: "Technical comments:"**
"It was hard to compare Figures 5a and 6, especially on the screen. As a suggestion, I wonder if these could be combined into a single plot or made into a single figure on the same page to make the comparison easier."

**Response:** We have done this.

**Changes in manuscript:** Figure 6 combined with Figure 5.

**Comment:** "The colour scale used in the panels of Figures 9 and also of Figure 11 differ. While the colour scales used aid the comparison of the spatial patterns, it makes comparison of the the absolute magnitudes harder. It might be better if a common scale could be adopted, with sufficient dynamic range and discretisation."

**Response:** When plotting these figures for the paper we gave some thought to the same issue raised by the referee and tried some plotting options. It was eventually a trade-off between keeping the same scale in all the plots and losing details (i.e. dynamic range). In our opinion, keeping a common scale would deteriorate the visibility of detailed gradients in the deposition velocity contours obtained using the two-layer reactivity model or would require nonlinear scaling values. Our thinking is to retain the current format of these figures.

**Changes in manuscript:** None.

[revised manuscript text omitted]

---

## Author Response (AR2)

**Reply by the authors to co-editor's comments**

Manuscript # acp-2016-844: "An improved parameterisation of ozone dry deposition to the ocean and its impact in a global climate-chemistry model" by Ashok K. Luhar et al.

5

10

20

25

We thank the co-editor, Dr Laurens Ganzeveld, for his comments. They have been particularly useful in addressing some loose ends in the paper. In the following, we provide a response to his comments. The reference to the changes made in the manuscript points to those in the 'Track Changes' version of the paper.

**Comment:** "I already wanted to provide this editors comment earlier so that you could have potentially included some of these in your response to the reviewer (editors) comments and for the revision of your paper. Unfortunately due to the workload this further feedback just comes after you have provided your

15 response and revision.

Some of these comments from my side also result from some interesting discussions I have had last week visiting some colleagues who also work on ocean-atmosphere exchange processes, in particular O3 and still want to introduce some of the points that come out of those discussions but also want to provide feedback on a couple of issues that already came across reading your ms after its initial submission and add up to the comments provided by the two reviewers."

**Response:** No problem. Thanks.

Changes in manuscript: None.

Comment: "Pp2; line 30: it is Wesely"

Changes in manuscript: Corrected.

**Comment:** "Pp3; line 11: I would leave out here the reference to Ganzeveld et al. 2009 since there we actually introduced the more mechanistic model. The one study that was discussed in that paper was the Kuhlmann et al., 2003 reference."

**Changes in manuscript:** The reference to Ganzeveld et al. (2009) has been removed.

**Comment:** "Pp4, line 25 and beyond: "but a detailed testing against deposition data was not reported, presumably because of the lack of suitable data at the time"

I don't agree with that statement. In the 2009 study we actually show a comparison of the simulated VdO3 and O3 dry deposition fluxes for the measurements that were available that time (Table 2 in Ganzeveld et al., 2009) and where we also evaluated how the comparison improved including the

potential role of DOM-O3 chemistry. You are correct that we couldn't include yet all the results reported in the Helmig et al. 2012 paper that you have applied in your paper but so might like to revise this statement."

**Changes in manuscript:** We agree with the co-editor. The last paragraph on Page 4 has been modified to incorporate his suggestion.

5 Comment: "Pp6: lne 21: "The deposition velocity is uniformly distributed to all model levels contained within the atmospheric boundary layer". Reading over this statement it interprets that the removal rate in the surface layer is smaller than the overall deposition rate or is the same deposition velocity used in all layers representing the boundary layer? and is this approach applied to potentially correct for issues on the representation of vertical transport?"

**Response:** Our statement could have been written a bit better. The way the present version of UKCA is setup, a first order loss rate is calculated as the deposition velocity divided by the height of the atmospheric boundary layer (ABL), and this loss rate is applied at each model level contained within the ABL. A grid-box mean loss rate at each model level within the ABL is then obtained by weighting the individual loss rates with the fractions of the surface types present in the grid box. A more appropriate method would be to divide the deposition velocity by the height of the lowest model level (or by the height of the surface layer) and apply the loss rate only at that height. According to the information provided by the U.K. Met Office, who are a UKCA developer, there are legacy reasons as to why the first method of loss calculation is used, and these are in no way related to any potential issues to do with the representation of vertical transport. The U.K. Met Office has tested a version of UKCA that has dry deposition loss done from the lowermost model level only and their results show that compared to the default case where deposition is applied within the full ABL, deposition fluxes change only marginally, if at all, whereas near-surface concentrations of ozone improve somewhat in comparison to measurements. We plan to explore this point further in the future.

**Changes in manuscript:** The last paragraph on Page 6 has been modified to better explain how the deposition velocity is used to determine the corresponding loss rate in the model (lines 25–28 on Page 6).

**10**

**Comment: "Pp 12, line 26: "This is not as..."**

Changes in manuscript: Corrected.

Comment: "Pp 16: lines 25 and beyond; in this short discussion on potential issues involved in the role of the reactivity at this point, but actually also further throughout the paper, you only elude on the role
15 of the Iodide whereas the total reactivity might also be strongly dependent on for example some of the DOM-O3 chemistry as discussed in more detail in the Ganzeveld et al., 2009 paper. In the revised paper you now mention explicitly that you have excluded in your study the role of organic compounds in your study. Still I think it would be worthwhile to shortly discuss how inclusion of this feature could further

affect the presented modelling results also in comparison to the 1-layer scheme. This chemistry could be especially relevant in the coastal zones but also in regions of strong upwelling with ocean productivity, in some of the colder waters of the Arctic but also in the southern oceans with cold conditions and where your analysis mainly makes a point about seeing there the largest differences between the 1- and

2 layer approach." 5

> **Response:** We provide a short discussion on this in the revised paper, incorporating a comment on relevant findings by Chang et al. (2004) and Ganzeveld et al. (2009).

> **Changes in manuscript:** The 2nd paragraph on Page 17 has been modified and new material added (lines 3–19 on Page 17).

Comment: "Pp16: I noticed that you removed the statement indicating that in the 2009 study we applied the nitrate as a proxy to indeed infer the Iodide concentrations (also since by that time the SST dependence was not yet presented). It seems that you have done so also because of the demonstrated

- stronger correlation between SST and Iodide. However, I think that it is actually important to explicitly 10 mention this essential difference in the inversion of the Iodide fields comparing the deposition schemes and modelling results. I also agree with one of the reviewers suggestion that it would have been interesting to potentially compare the impact of the differences in inferring Iodide (also including a run with the 2-layer model with Iodide based on nitrate) also because, despite the apparent stronger
- correlation between SST and Iodide, this feature of getting realistic oceanic biogeochemical boundary 15 conditions for these deposition calculations seems to be a remaining essential challenge also due to limited observations."

Response: The statement indicating that Ganzeveld et al. (2009) used oceanic surface nitrate concentration as a proxy for iodide concentration has been reinstated. We need more observations to relate iodide with other parameters of the ocean biogeochemistry (such as nitrate). Chance et al. (2014) provide a good account and analysis of available data to examine iodide dependencies, and perhaps these data could be further analysed for better parameterisations. Of course, we also need more comprehensive deposition velocity measurements so that the impact of changes in iodide parameterisations as well as those of any additional chemical reactions could be discerned and quantified.

Changes in manuscript: The sentence "Ganzeveld et al. (2009) used oceanic surface nitrate concentration as a proxy for iodide concentration" is added subsequent to Eq. (11). In addition, a sentence in the last paragraph of Conclusions added (lines 18–20, Page 22).

Comment: "The following comment also relates to one of the comments made by reviewer #2. Referring to your discussion on the global and oceanic deposition budget. I noticed that with your 20 global model simulations with an oceanic ozone deposition sink which is 12% of the global sink, you arrive at a flux of about 80 TgO3 yr-1. You discuss that this for the 1-layer model it would be about 19% and for the fixed rc approach about ~21% or so. This would imply a global oceanic O3 deposition

budget of 180 TgO3 yr-1 which is substantially smaller than the 300 TgO3 yr-1 global oceanic O3 deposition term presented by Ganzeveld et al. (2009). Is raises a question how to appreciate these differences between the 1- and 2- layer oceanic O3 deposition model recognizing already the fact that there seems to be like a ~factor 2 difference among such global chemistry-climate models inferring this

5 ozone budget term. It also comes back to my remark about the deposition approach in your model using the vd uniformely distributed over all boundary layers. Rather than evaluating the O3 dry deposition velocities, it would be useful to also evaluate the O3 dry deposition fluxes by comparison with the observations. Have you conducted such an evaluation??"

**Response:** The figures mentioned by the co-editor are correct. There is quite a range of modelled total ozone dry deposition reported in the literature, and some of these ranges are mentioned in the 1st paragraph of Section 7.2. The range cited by Ganzeveld et al. (2009) is 600–1000 Tg yr-1. As mentioned in our paper, our deposition budget values are on the lower end of the values reported based on other models. We think the main reason for that is that UKCA underestimates tropospheric ozone concentrations, which results in an underestimation of the ozone dry deposition flux at the surface. However, note that the calculated deposition velocity is not influenced by the predicted ozone concentration, and hence the relative (not absolute) dry deposition budget contributions given in our paper in terms of percentages for the various deposition schemes should remain more or less the same even if the model was predicting the ozone concentrations fine. The two-layer scheme agrees the best with the Helmig et al. deposition velocity data, and we emphasize the relative change in oceanic dry deposition when this scheme is used. A model that predicts tropospheric ozone correctly and includes a correct description of deposition velocity should yield a realistic value of the total deposition budget. We agree that it would also be useful to evaluate the ozone dry deposition flux, but for that to be meaningful we first need to fix the issue of the underestimation of tropospheric ozone concentration in the model because such a flux is proportional to the concentration given that the deposition velocity would change little from the present values. In this regard, we think that our approach of using deposition velocity uniformly distributed within the boundary layer is not a problem (see our response to the related comment above).

[revised manuscript text omitted]